# PseuZO: Pseudo-Zeroth-Order Algorithm for Training Deep Neural Networks

**Pengyun Yue[1,*], Xuanlin Yang[1,5,*], Mingqing Xiao[1,4], Zhouchen Lin[1,2,3,†]**

[1]State Key Lab of General AI, School of Intelligence Science and Technology, Peking University
[2]Institute for Artificial Intelligence, Peking University
[3] Pazhou Laboratory (Huangpu), Guangzhou, Guangdong, China
[4] Microsoft Research Asia
[5] Zhongguancun Academy
yuepy@pku.edu.cn,xuanlinyang@stu.pku.edu.cn,{mingqing_xiao,zlin}@pku.edu.cn

## Abstract

Zeroth-order Optimization (ZO) has received wide attention in machine learning, especially when computing full gradient is expensive or even impossible. Recently, ZO has emerged as an important paradigm for memory-efficient fine-tuning of large language models (LLMs), circumventing the memory overhead of backpropagation. However, existing ZO gradient estimators exhibit dimension-dependent variance scaling as $\Theta(d)$, leading to dimension-dependent convergence rates without further assumptions on the objective function, which is prohibitive for large-scale LLM parameters. To address this problem, we present a Pseudo-Zeroth-Order (PseuZO) framework for optimizing composite objective functions, especially large-scale models: $\min_{\mathbf{x} \in \mathcal{X}} \mathcal{F}(\mathbf{x}) = \mathbb{E}_{\mathbf{z}} g \circ h(\mathbf{x}; \mathbf{z})$, where $h$ represents complex, high-dimensional representations and $g$ is a task-specific loss. While existing zeroth-order methods estimate gradients with final loss functions, our PseuZO algorithm estimate the Jacobian matrix of $h(\mathbf{x})$ with the model output $\mathbf{o} = h(\mathbf{x})$, and the gradient of the loss function on model output $\mathbf{e} = \nabla_{\mathbf{o}} g(\mathbf{o})$, and apply exponential moving average on Jacobian estimators to reduce the variance. Moreover, we use the sliding window technique to reduce memory costs. Our algorithm achieves an $O\left(\max\left\{\alpha_1 L \epsilon^{-2}, \alpha_1 L \sigma_2^2 \epsilon^{-4}\right\}\right)$ convergence rate, where $\alpha_1$ is the effective dimension of $\mathcal{F}$. Experimental results demonstrate that PseuZO outperforms MeZO and MeZO-SVRG in classification, multiple choice and generation tasks in both full-parameter and PEFT fine-tuning settings by boosting convergence in the early stages of training. For instance, under the same computation time, with respect to SST2 task, PesuZO gets 9.8% higher accuracy than MeZO (91.2% v.s. 82.4%). With the sliding window technique, our PseuZO achieves $70\% \sim 80\%$ memory reduction compared to FO-SGD for different model sizes as PseuZO only introduced a small dimension-independent memory overhead, which enables efficient scaling of the model size. The code is available at https://github.com/YangBigMn/PseuZO.

## 1 Introduction

Zeroth-order optimization [45, 20, 36] has served as a core technique for problems where gradient calculations are impractical. These methods rely solely on function evaluations, making them uniquely suited for black-box scenarios like adversarial attacks [6, 59], reinforcement learning [8, 14], and hyperparameter tuning [23, 37]. Compared to first-order and higher-order optimization methods,

---

*Equal contribution.
†Corresponding author.

39th Conference on Neural Information Processing Systems (NeurIPS 2025).

their key strength lies in avoiding gradient computations—a critical advantage when optimizing complex systems where automatic differentiation is infeasible or prohibitively expensive. For modern deep neural networks, this approach significantly reduces memory demands by eliminating backpropagation's computational and memory cost. Recent advances have reinvigorated zeroth-order methods as practical tools for deep neural network training, particularly in resource-constrained environments where traditional optimization strategies struggle to scale.

Large language models (LLMs) represent the pinnacle of deep neural network architectures, achieving state-of-the-art performance across diverse language understanding and generation tasks [48, 50, 5, 16]. The standard paradigm of pretraining on web-scale corpora followed by task-specific fine-tuning has become ubiquitous, enabling these models to adapt to specialized domains. However, conventional first-order fine-tuning approaches employing full-parameter optimization through backpropagation face critical scalability barriers: the memory overhead for storing optimizer states and activation values grows with model parameters and context length, becoming prohibitive for large-scale models. This challenge has driven the emergence of parameter-efficient fine-tuning (PEFT) techniques [31, 25, 35, 55] that strategically update only subsets of model weights. Another way to reduce memory costs is using memory-efficient zeroth-order optimization (MeZO)[34]. MeZO introduces a paradigm shift by operating purely through forward-pass evaluations. By eliminating backpropagation while maintaining competitive task adaptation capability, MeZO addresses the dual requirements of memory conservation and optimization stability in resource-constrained scenarios, enabling the deployment of massive LLMs in practical applications.

While zeroth-order optimization provides a gradient-free alternative for fine-tuning large neural networks, its practical adoption faces fundamental limitations. Zeroth-order algorithms exhibit catastrophic performance when training from scratch. Even for fine-tuning tasks, the convergence rate of classical zeroth-order methods scales linearly with parameter dimension $d$, becoming prohibitive for modern architectures where $d$ routinely exceeds $10^{10}$. This dimension dependence persists in practice even when leveraging low-effective dimensionality theories [56] or the sparsity of the model structure [24].

In this paper, we propose a Pseudo-Zeroth-Order (PseuZO) framework for optimizing composite objective functions:

$$\min_{\mathbf{x} \in \mathcal{X}} \mathbb{E}_{\mathbf{z} \sim \mathcal{D}} g \circ h(\mathbf{x}; \mathbf{z}). \tag{1}$$

This problem formulation is prevalent in deep neural network training, where $h$ represents complex, high-dimensional representations and $g$ is a task-specific loss. Unlike traditional zeroth-order methods, PseuZO methods estimate the Jacobian matrix of $h(\mathbf{x})$ with the model output $\mathbf{o} = h(\mathbf{x})$. Compared to the value of the composite function $g \circ h$, the output of $h$ provides more information about the function $h$. We also apply exponential moving average on Jacobian estimators to reduce the variance. Finally, we obtain the gradient estimator with the Jacobian estimator and the gradient of the loss function on model output $\mathbf{e} = \nabla_{\mathbf{o}} g(\mathbf{o})$, which can often be computed explicitly. We demonstrate the efficacy of PseuZO methods both theoretically and empirically. In theory, we prove that PseuZO method finds an $\epsilon$-stationary point with $O\left(\max\left\{\alpha_1 L \epsilon^{-2}, \alpha_1 L \sigma_2^2 \epsilon^{-4}\right\}\right)$ function evaluations, where $\alpha_1$ is the effective dimension of the objective function $\mathcal{F}$. In our experiments, PseuZO not only converges faster, but also attains a precision improvement of up to 9.8% compared to MeZO. With the sliding window technique, PseuZO only needs a small dimension-independent extra memory overhead compared to MeZO, which enables efficient scaling of the model size. Additionally, we incorporate PseuZO with LoRA and prefix-tuning to show that PseuZO is also compatible with PEFT techniques.

We summarize our main contributions below:

1. We propose PseuZO optimization framework, which uses the differentiation of model outputs to compute a stochastic Jacobian estimator, and apply exponential moving average to reduce the variance. In practice, we use the sliding window technique to reduce memory costs.

2. We proved that the convergence rate of PseuZO method is not explicitly dependent on the parameter size. We theoretically prove that our PseuZO optimization method finds an $\epsilon$-stationary point in $O\left(\max\left\{\alpha_1 L \epsilon^{-2}, \alpha_1 L \sigma_2^2 \epsilon^{-4}\right\}\right)$ function evaluations, where $\alpha_1$ is the effective dimension.

3. We conduct solid and comprehensive experiments which show that PseuZO outperforms ICL and MeZO across multiple tasks, including classification, multi-classification and generation in terms of convergence speed. With the sliding window technique, PseuZO only shows a small parameter-size-independent memory overhead compared to MeZO for instruction fine-tuning tasks. Moreover, we find that PseuZO is compatible with PEFT like LoRA and prefix-tuning, and results show that PseuZO+PEFT also outperforms MeZO+PEFT across classification, multi-classification and generation tasks in terms of convergence speed.

## 2    Related work

**Zeroth-order optimization**. Zeroth-order optimization [45, 20, 36] has been widely studied in the field of machine learning, and has been used in black-box optimization [22, 6, 59], adversarial attacks [9, 49], etc. Most zeroth-order methods are designed based on first-order [36] or higher-order methods [56], and are often $d$ times slower where $d$ is the dimension of the problem. To mitigate the curse of dimensionality, several works proposed effective dimension [56, 34], and characterize the convergence rate with the effective dimension of the problem. Many studies also consider reformulating the neural network at a relatively small scale to solve a simpler optimization problem [30, 47, 7], and then utilize block coordinate descent (BCD) [6] or ADMM [33] without the need for gradients. Recently, MeZO [34] successfully applied zeroth-order optimization to fine-tuning extremely large language models by efficiently estimating gradients in memory. After that, many works attempt to improve the performance of MeZO by reducing variance [19] or introducing estimated second-order information [60]. In the research of Spike Neural Network (SNN), [54] used the model output to estimate the Jacobian matrix, but their work was designed from biological applicability, and was not able to save memory costs. Inspired by [54], we designed our PseuZO optimization framework.

**Memory-efficient backpropagation**. As LLMs are typically fine-tuned by first-order algorithms like SGD [42] and Adam [28], many new methods or techniques have been proposed to solve the memory overhead problem, e.g. sparsifying gradients [46, 53] and quantization [15]. Other useful techniques to save memory for activation values or optimizer states like Gradient Checkpoint [18], Flash Attention [12] and Zero Redundancy Optimizer (ZeRO) [41, 39, 40]. However, these methods either sacrifice precision or require more computation time.

**Gradient-free adaptation of LLMs**. Language models can understand language and learn to communicate with humans after the pre-training phase. They can then generalize to tasks without training and this form adaptation that requires appropriate prompt designs is called in-context learning (ICL). Another paradigm is to estimate first-order or second-order information only using inference. Besides MeZO estimating first-order information by two forward processes, HiZOO leverages three forward processes to estimate second-order information considering heterogeneous curvatures across different parameter dimensions [60].

## 3    Pseudo-Zeroth-order algorithm framework

### 3.1    Zeroth-order algorithms

Zeroth-order algorithms are a class of optimization algorithms that do not require the computation of gradients. Instead, they rely on noisy function value oracles to update the parameters. This makes them useful for problems where the computation of gradients is expensive or infeasible. Typically, zeroth-order methods use noisy function values to generate gradient estimators. Suppose the objective function is $f : \mathbb{R}_d \to \mathbb{R}$. Two most common gradient estimators are RGE [36] and CGE [1]:

$$\text{RGE}: \tilde{\nabla}_\mu f(\mathbf{x}) = \frac{1}{q} \sum_{i=1}^{q} \frac{f(\mathbf{x} + \mu \boldsymbol{\xi}_i) - f(\mathbf{x})}{\mu} \boldsymbol{\xi}_i, \ \text{CGE}: \tilde{\nabla}_\mu f(\mathbf{x}) = \frac{1}{d} \sum_{i=1}^{d} \frac{f(\mathbf{x} + \mu \mathbf{e}_i) - f(\mathbf{x})}{\mu} \mathbf{e}_i,$$

$$(2)$$

where $\boldsymbol{\xi}_i \sim N(\mathbf{0}, \mathbf{I})$ and $\{\mathbf{e}_i\}$ is a set of standard basis vector. When $\mu \to 0$, CGE become the full gradient, and RGE tends to

$$\hat{\nabla} f(\mathbf{x}) = \frac{1}{q} \sum_{i=1}^{q} \langle \nabla f(\mathbf{x}), \boldsymbol{\xi}_i \rangle \boldsymbol{\xi}_i. \tag{3}$$

This is an unbiased estimation of $\nabla f(\mathbf{x})$, with variance $\Theta\left(\frac{d}{q}\right)$. Without further assumptions on objective functions, optimizing with RGE or CGE needs $d$ times more zeroth-order oracles than optimizing with first-order methods using gradient oracles, severely limiting their applicability in high-dimensional scenarios. However, if the objective function has low effective dimension or sparsity structures, zeroth-order methods can achieve a faster convergence rate [56, 60].

## 3.2 Pseudo-zeroth-order algorithms

In this paper, we study the composite optimization problem:

$$\min_{\mathbf{x} \in \mathcal{X}} \mathcal{F}(\mathbf{x}) := \mathbb{E}_{\mathbf{z}} g(h(\mathbf{x}; \mathbf{z})), \quad \text{where } \mathcal{X} \subseteq \mathbb{R}^{d_{\mathrm{p}}}, \tag{4}$$

where $h : \mathcal{X} \to \mathcal{H} \subseteq \mathbb{R}^{d_{\mathrm{out}}}$ defines the representation mapping and $g : \mathcal{H} \to \mathbb{R}_+$ is a loss function. In machine learning, the representation mapping $h : \mathbb{R}^{d_{\mathrm{p}}} \to \mathbb{R}^{d_{\mathrm{out}}}$ is typically parameterized as a neural network $h(\mathbf{x}; \mathbf{z})$, where $\mathbf{x}$ denotes the learnable parameters and $\mathbf{z} \sim \mathcal{D}$ denotes the stochastic input data. The exact stochastic gradient admits the theoretical decomposition:

$$\nabla_{\mathbf{x}} \mathcal{F}(\mathbf{x}) = \mathbb{E}_{\mathbf{z} \sim \mathcal{D}} \Big[ \underbrace{\mathbf{J}_h^\top(\mathbf{x}; \mathbf{z})}_{\substack{\text{Stochastic Jacobian} \\ d_{\mathrm{out}} \times d_{\mathrm{p}}}} \cdot \underbrace{\nabla_{\mathbf{h}} g(h(\mathbf{x}; \mathbf{z}))}_{\substack{\text{Upstream gradient} \\ d_{\mathrm{out}} \to \mathbb{R}}} \Big]. \tag{5}$$

The implementation of stochastic first-order methods requires efficient computation of $\nabla_{\mathbf{x}} \mathcal{F}(\mathbf{x})$ via backpropagation through the compositional structure $g \circ h$. While the outer loss $g$ admits tractable gradient computation ($\nabla_{\mathbf{h}} g$ is typically closed-form), the inner mapping's Jacobian $\mathbf{J}_h(\mathbf{x}; \mathbf{z})$ becomes computationally intractable for deep nonlinear parameterizations.

Unlike standard zeroth-order (ZO) methods that solely utilize function evaluations of $g \circ h$, our key insight stems from the asymmetric differentiability inherent in composite optimization: While acquiring exact gradients through the inner mapping $h(\mathbf{x}; \mathbf{z})$ remains computationally prohibitive due to computational constraints or non-differentiable operators, the gradient of the outer function $g$ can be explicitly or efficiently computed.

Our method exploits composite structure's asymmetric differentiability:

- **Outer gradient**: Closed-form $\mathbf{e} = \nabla_{\mathbf{o}} g(\mathbf{o})$ for $\mathbf{o} = h(\mathbf{x}; \mathbf{z})$;

- **Inner estimation**: Zeroth-order approximation for $h$'s Jacobian.

The PZO gradient estimator combines both components:

$$\nabla_{\mu}^{\mathrm{PZO}} \mathcal{F}(\mathbf{x}; \mathbf{z}) = \boldsymbol{\xi} \left( \frac{h(\mathbf{x} + \mu \boldsymbol{\xi}; \mathbf{z}) - h(\mathbf{x}; \mathbf{z})}{\mu} \right)^\top \mathbf{e}, \tag{6}$$

where $\boldsymbol{\xi} \sim N(\mathbf{0}, \mathbf{I})$. As $\mu \to 0$, this converges to $\nabla^{\mathrm{PZO}} \mathcal{F}(\mathbf{x}; \mathbf{z}) = (\boldsymbol{\xi}^\top \mathbf{J}_h^\top \mathbf{e}) \boldsymbol{\xi} = \boldsymbol{\xi} \boldsymbol{\xi}^\top \mathbf{J}_h^\top \mathbf{e}$, which is an unbiased estimation of $\nabla \mathcal{F}(\mathbf{x})$. To reduce the variance introduced by the random vector $\boldsymbol{\xi}$, we propose a momentum-accelerated variant through exponential smoothing:

$$\mathbf{A} = (1 - \lambda) \left( \frac{h(\mathbf{x} + \mu \boldsymbol{\xi}; \mathbf{z}) - h(\mathbf{x}; \mathbf{z})}{\mu} \right) \boldsymbol{\xi}^\top + \lambda \mathbf{A}, \tag{7}$$

$$\hat{\nabla}_{\mu}^{\mathrm{PZO}} \mathcal{F}(\mathbf{x}; \mathbf{z}) = \mathbf{A}^\top \mathbf{e}. \tag{8}$$

The full algorithm is shown in Algorithm 1.

**Remark 1.** *The gradient estimator of PseuZO $\hat{\nabla}_{\mu}^{PZO} \mathcal{F}(\mathbf{x}; \mathbf{z}) = \mathbf{A}^\top \mathbf{e}$ theoretically outperforms MeZO and its momentum variant in terms of both bias and variance, as detailed in Appendix D. PseuZO has smaller variance compared to MeZO and MeZO with momenetum, and MeZO has an extra bias term caused by the curvature of $g$. These theoretical advantages demonstrate the effectiveness of the exact outer gradient and exponential smoothing of Jacobian estimations.*

---

**Algorithm 1** Matrix-based PseuZO Algorithm

---

**Require:** Momentum factor $\beta \in (0, 1)$, smoothing coefficient $\mu > 0$, max iterations $T$, initial point $\mathbf{x}_0$

**Ensure:** Gradient estimate $\hat{\nabla}_\mu^{\text{PZO}} \mathcal{F}(\mathbf{x}; \mathbf{z})$

1: Initialize momentum buffer $\mathbf{A}_{-1} \leftarrow \mathbf{0} \in \mathbb{R}^{d_{\text{out}} \times d_{\text{p}}}$
2: **for** $t = 0$ to $T$ **do**
3:      Sample random vector $\boldsymbol{\xi}_t \sim \mathcal{N}(\mathbf{0}, \mathbf{I})$
4:      Compute forward difference $\Delta \mathbf{o}_t \leftarrow \frac{h(\mathbf{x}_t + \mu \boldsymbol{\xi}_t; \mathbf{z}_t) - h(\mathbf{x}_t; \mathbf{z}_t)}{\mu}$
5:      Receive noisy Jacobian estimator $\mathbf{B}_t = \Delta \mathbf{o}_t \boldsymbol{\xi}_t^\top$
6:      Momentum update:
7:        $\mathbf{A}_t \leftarrow (1 - \lambda_t) \mathbf{B}_t + \lambda_t \mathbf{A}_{t-1}$            ▷ Exponential moving average
8:      Compute outer gradient $\mathbf{e}_t \leftarrow \nabla_{\mathbf{o}} g(h(\mathbf{x}; \mathbf{z}_t))$
9:      Gradient projection: $\hat{\nabla}_\mu^{\text{PZO}} \mathcal{F}(\mathbf{x}_t; \mathbf{z}_t) \leftarrow \mathbf{A}_t^\top \mathbf{e}_t$
10:     Update $\mathbf{x}$: $\mathbf{x}_{t+1} \leftarrow \mathbf{x}_t - \eta \hat{\nabla}_\mu^{\text{PZO}} \mathcal{F}(\mathbf{x}; \mathbf{z}_t)$
11: **end for**

---

### 3.3 Convergence of PseuZO algorithm

In this subsection, we deviate from the exact realization of stochastic Jacobian estimators, and assume that the PseuZO algorithm receives a noisy Jacobian estimator in each step. Under this more general setting, we prove the convergence of PseuZO method. We first list some assumptions which are necessary for our analysis and has been widely adopted in research works on optimization:

**Assumption 2.** $\mathcal{F}$ *has continuous Hessian matrices* $\mathbf{H}(\mathbf{x})$, *and satisfy the following equations:*

$$\|\mathbf{H}(\mathbf{x})\|_{\text{op}} \leq L, \quad \text{tr}\left(\mathbf{H}(\mathbf{x})\right) \leq \alpha_1 L, \tag{9}$$

*where* $\alpha_1$ *is the effective dimension of* $\mathcal{F}$. *In the worst case,* $\alpha_1 = d_p$.

**Assumption 3.** *Denote* $\hat{\nabla}\mathcal{F}(\mathbf{x}_t) = \nabla_{\mathbf{x}} g(h(\mathbf{x}_t; \mathbf{z}_t))$. *The randomness of the Jacobian estimator* $\mathbf{B}_t$ *can be decoupled into the following parts:*

$$\mathbf{B}_t = (\mathbf{J}_t + \mathbf{D}_t)\mathbf{N}_t + \mathbf{M}_t, \tag{10}$$

*where* $\mathbf{J}_t$ *is the true Jacobian matrix of* $h$ *at* $\mathbf{x}_t$, *and:*

- $\mathbf{N}_t$ *represents the randomness introduced by the PZO Jacobian estimator:*

$$\mathbb{E}\mathbf{N}_t = \mathbf{I}, \quad \mathbb{E}\|\mathbf{N}_t^\top \mathbf{a}\|_{\mathbf{M}}^2 \leq 3\text{tr}\left(\mathbf{M}\right) \cdot \|\mathbf{a}\|^2, \tag{11}$$

     *where* $\mathbf{a}$ *is an arbitrary vector.*

- $\mathbf{D}_t$ *represents the randomness of data:*

$$(\mathbf{J}_t + \mathbf{D}_t)^\top \mathbf{e}_t = \hat{\nabla}\mathcal{F}(\mathbf{x}_t), \quad \mathbb{E}\mathbf{D}_t = 0, \quad \mathbb{E}\|\mathbf{D}_t^\top \mathbf{e}_t\|^2 \leq \sigma_2^2; \tag{12}$$

- $\mathbf{M}_t$ *represents the noise introduced by the two-point estimation of the function value, controlled by* $\mu$:

$$\mathbb{E}\|\mathbf{M}_t^\top \mathbf{e}_t\|^2 \leq \frac{\mu^2 L_h^2 L_g^2}{2}(d_p + 6)^3. \tag{13}$$

According to the analysis on the two-point gradient estimators in [36], $\mathbf{B}_t = \left(\frac{h(\mathbf{x} + \mu \boldsymbol{\xi}; \mathbf{z}) - h(\mathbf{x}; \mathbf{z})}{\mu}\right) \boldsymbol{\xi}^\top$ satisfies Assumption 3, where $\boldsymbol{\xi} \sim N(\mathbf{0}, \mathbf{I})$. Now, we propose the informal version of our convergence theorem in Theorem 4. For the formal version and the proof of Theorem 4, please refer to the Appendix C.

**Theorem 4** (Informal). *Under Assumption 2 and 3, if* $\mu$, $\epsilon$ *and* $\beta_t$ *satisfy certain conditions, Algorithm 1 finds an* $\epsilon$-*stationary point with* $O\left(\max\left\{\alpha_1 L\epsilon^{-2}, \alpha_1 L\sigma_2^2 \epsilon^{-4}\right\}\right)$ *function value computations.*

**Remark 5.** *The convergence rate of Algorithm 1 is* $\alpha_1$ *times slower than the standard SGD algorithm, where* $\alpha_1$ *is not explicitly dependent on the dimension of the problem.*

### 3.4 Sliding window-based PseuZO algorithm

As it is memory inefficient to store such a large-scale tensor $\mathbf{A}_t$ (outer product of Gaussian noise and an output tensor) introduced in PseuZO, we propose several techniques to reduce the memory cost and obtain Sliding Window-based PseuZO shown in Algorithm 2.

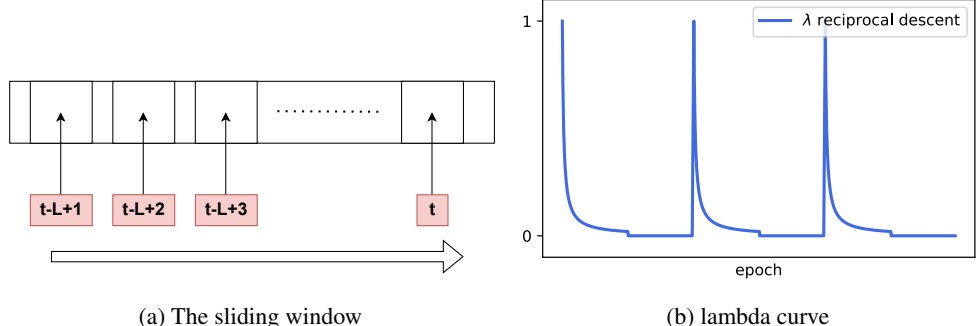

(a) The sliding window             (b) lambda curve

Figure 1: (a) is a schematic diagram of the sliding window with length $L$. We can obtain the corresponding coefficients of all sliding window units by expanding the iterative $L$ times. (b) is a $\lambda$ variation curve with three cycles. Each cycle means a restart operation and when $\lambda$ is close to zero, we set $\lambda = 0$ to reduce time consumption.

**Sliding window technique**. To tackle the memory overhead of PseuZO, we use a sliding window to store random seeds and output tensors in the last few steps. When the sliding window shown in Figure 1a is determined, the corresponding coefficient for each unit in the sliding window can be obtained by expanding the EMA formula. On the one hand, we can resample the same Gaussian noise $\xi$ with the same seed so there is no need to store these random vectors [34]. On the other hand, as $0 < \lambda < 1$, especially when $\lambda$ is smaller, the weight for older information is close to zero in the EMA step of PseuZO method. Therefore, we truncate outdated information with the sliding window technique.

**Changing the storage target**. However, storing several output tensors with a shape of the vocabulary size is still unacceptable, as the vocabulary size is generally large among LLMs. So we choose to store the last hidden state with a much smaller size and trace its path to the final loss to obtain its gradient without excessive memory overhead. For OPT1.3B, the last hidden state (i.e. the input tensor for **lm_head**) size is 2048 while the vocabulary size is larger than 50000 [58]. Thus we can further reduce memory overhead caused by the sliding window.

**Periodic dynamic changing of** $\lambda$. To meet the convergence requirements, $\lambda$ needs to gradually decrease to zero. We take advantage of a restart operation to further boost convergence and thus $\lambda$ is designed in a periodic changing manner as shown in Figure 1b. When $\lambda$ approaches zero, PseuZO gradually degenerates to ZO method and thus we directly transform to MeZO instead. If we consistently use PseuZO throughout the entire process, the time cost is roughly $2\times$ that of MeZO. However, with appropriate design for $\lambda$, the time cost can be reduced to almost the same as MeZO.

## 4 Experiments

In this section, we evaluate Sliding Window-based PseuZO on a variety of typical fine-tuning tasks by comparing performance against MeZO [34], MeZO-SVRG [19], HiZOO-L [60] and memory overhead against MeZO-SVRG as well as FO-SGD. Experimental results show that: 1) The peak memory usage of PseuZO is significantly smaller than MeZO-SVRG and FO-SGD; 2) Through the sliding window technique, with a fixed memory overhead that is independent of the model size compared to MeZO, PseuZO has much better performance than MeZO, MeZO-SVRG and HiZOO-L. 3) PseuZO is also compatible with PEFT techniques like LoRA [25] and prefix-tuning [31].

**Setup.** We implement PseuZO, MeZO-SVRG and HiZOO-L in the MeZO framework with appropriate adjustment for fair comparison. We conduct comprehensive experiments in various tasks on large auto-regressive language models like opt-1.3B [58] and the same prompt design as MeZO is utilized which is effective and fair for comparison for various datasets including GLUE [52] and SuperGLUE

---

**Algorithm 2** Sliding Window-based PseuZO Algorithm

---

**Require:** Momentum factor formula $\lambda(t) \in (0, 1)$, smoothing coefficient $\epsilon > 0$, max iterations $T$, initial point $\theta_0$, sliding window length $L$, coefficients $\{u_k\}_{k=1}^L$
**Ensure:** Gradient estimate $\hat{\nabla}_\epsilon^{\text{PZO}} \mathcal{F}(\mathbf{x}; \mathbf{z})$
1: Initialize the sliding window as a deque $D(\text{maxlen}=L)$
2: **for** $t = 0$ **to** $T$ **do**
3:     Update $\lambda \leftarrow \lambda(t)$
4:     Compute coefficients $u_k \leftarrow \lambda^{k-1}(1 - \lambda)$
5:     Sample random seed $s_t$ and corresponding vector $\boldsymbol{\xi}_t \sim \mathcal{N}(\mathbf{0}, \mathbf{I}; s_t)$
6:     Compute forward difference $\Delta \mathbf{o}_t \leftarrow \frac{h(\mathbf{x}_t + \epsilon \boldsymbol{\xi}_t; \mathbf{z}_t) - h(\mathbf{x}_t; \mathbf{z}_t)}{\epsilon}$
7:     Compute outer gradient $\mathbf{e}_t \leftarrow \nabla_\mathbf{o} g(h(\mathbf{x}_t; \mathbf{z}_t))$
8:     Update sliding window $D$.append($s_t, \Delta \mathbf{o}_t$)
9:     Gradient projection initialization: $\hat{\nabla}_\epsilon^{\text{PZO}} \mathcal{F}(\mathbf{x}_t; \mathbf{z}_t) \leftarrow 0$
10:     **for** $k = 1$ **to** $L$ **do**                                      ▷ Iterate sliding window
11:         $s, \Delta \mathbf{o} \leftarrow D_k$
12:         Resample $\boldsymbol{\xi} \sim \mathcal{N}(\mathbf{0}, \mathbf{I}; s)$
13:         Accumulate $\hat{\nabla}_\epsilon^{\text{PZO}} \mathcal{F}(\mathbf{x}_t; \mathbf{z}_t) \leftarrow \hat{\nabla}_\epsilon^{\text{PZO}} \mathcal{F}(\mathbf{x}_t; \mathbf{z}_t) + u_k \langle \Delta \mathbf{o}, \mathbf{e}_t \rangle \boldsymbol{\xi}$
14:     **end for**
15:     Update $\theta$: $\theta_{t+1} \leftarrow \theta_t - \eta \hat{\nabla}_\epsilon^{\text{PZO}} \mathcal{F}(\theta_t; \mathbf{x}_t)$
16: **end for**

---

| Task
Task type | SST-2 | RTE | CB | BoolQ | WSC | WIC | MultiRC | COPA | ReCoRD | DROP |
|---|---|---|---|---|---|---|---|---|---|---|
| | | | | classification | | | | multiple choice | | generation |
| Zero-shot | 53.5 | 53.0 | 39.3 | 45.7 | 43.3 | 51.5 | 45.4 | 75.0 | 70.5 | 11.2 |
| ICL | 80.0 | 53.0 | 46.4 | 58.7 | 47.1 | 51.1 | 46.2 | 69.0 | **71.0** | 20.4 |
| MeZO-SVRG | 61.5 | 55.5 | 74.0 | 60.3 | 52.0 | 50.0 | 53.0 | 54.0 | 50.1 | 0.0 |
| MeZO (10K steps) | 82.4 | 54.3 | 76.0 | 60.7 | 51.0 | 50.9 | 54.9 | 74.0 | 57.6 | 20.3 |
| MeZO (20K steps) | 88.4 | 58.8 | 76.0 | 63.8 | 53.0 | 51.3 | 53.9 | 73.0 | 58.9 | 20.3 |
| HiZOO-L | 88.1 | 54.9 | 69.0 | 64.8 | 51.2 | 58.0 | 58.2 | 73.0 | 58.8 | 23.3 |
| PseuZO (10K steps) | **91.2** | 58.0 | **77.0** | 64.3 | **58.0** | 54.5 | 54.7 | **78.0** | 60.0 | 23.5 |
| PseuZO (20K step) | 90.7 | **63.3** | 75.0 | **67.0** | 57.0 | **59.7** | **60.6** | 76.0 | 60.9 | **24.5** |
| FO-SGD | 92.4 | 67.8 | 94.0 | 60.8 | 52.6 | 47.4 | 53.8 | 76.0 | 57.2 | 26.0 |

Table 1: Experiments on OPT-1.3B with 1024 training samples and 512 evaluation samples. When training, for WSC, CB and COPA, they have much less total samples and thus we set aside 100 evaluation samples and use the rest for training. The **bold** number represents the highest evaluation performance excluding FO-SGD.

[51] benchmarks. We run all experiments for **10K** steps and evaluate performance of the model every **2K** steps for HiZOO-L and MeZO-SVRG. In order to ensure that MeZO and PseuZO are sufficiently convergent, we run PseuZO and MeZO for **10K** and **20K** steps, respectively. We choose $K = 16$ as the batch size and randomly select 1024 samples for training and 512 samples for evaluation. All experiments are run on a single Nvidia A800 40GiB GPU.

| Task | SST-2 | RTE | CB | BoolQ | WSC | WIC | COPA |
|---|---|---|---|---|---|---|---|
| MeZO | 84.4 | 58.0 | 76.0 | 64.0 | 54.0 | 52.5 | 88.0 |
| PseuZO (ours) | **91.8** | **58.4** | **77.0** | **68.9** | **58.0** | **55.3** | **90.0** |

Table 2: Experiments on OPT-6.7B for PseuZO versus MeZO. The **bold** number represents the better evaluation performance.

| Task
Task type | SST-2 | RTE | CB | BoolQ | WSC | WIC | MultiRC | COPA | ReCoRD | DROP |
|---|---|---|---|---|---|---|---|---|---|---|
| | | | | _classification_ | | | | _multiple choice_ | | generation |
| MeZO+LoRA | 90.8 | **59.1** | 76.0 | 64.6 | 50.0 | **52.7** | **54.5** | 81.0 | 59.5 | 22.9 |
| PseuZO+LoRA | **91.2** | 58.8 | **79.0** | **65.8** | **51.0** | 51.8 | 54.3 | **83.0** | **60.2** | **25.7** |
| MeZO+prefix | 71.0 | 51.7 | 45.0 | 57.6 | **54.0** | 50.6 | 50.8 | 75.0 | 57.4 | 15.5 |
| PseuZO+prefix | **80.7** | **53.1** | **71.0** | **61.7** | 49.0 | **51.0** | **52.7** | **75.0** | **57.8** | **21.2** |

Table 3: Experiments on OPT-1.3B for comparison between MeZO+PEFT and PseuZO+PEFT. PEFT is either LoRA or prefix fine-tuning.

## 4.1 Auto regressive model performance

**PseuZO performs much better than MeZO, MeZO-SVRG and HiZOO-L in the classification, multiple choice and generation tasks** shown in Table 1. As MeZO-SVRG needs to traverse all samples to obtain the full-batch gradient [19], large numbers of iterations are required or it will have an expensive memory overhead. However, MeZO-SVRG still performs worse than PseuZO, especially for multiple choice and generation tasks with a small batch size and a large full batch size. For HiZOO-L [60], it not only requires three forward propagations, but also requires low-rank processing of matrix parameters, which increases the time overhead. In fact, running HiZOO-L and MeZO-SVRG for only 10K steps takes much longer than running PseuZO for 20K steps. As shown in Appendix D, PseuZO takes advantage of the small bias and variance to achieve fast convergence, further reducing the gap with FO-SGD and even outperforms FO-SGD in many tasks.

**PseuZO is also compatible with other memory efficient techniques, like LoRA and prefix tuning**, and the corresponding results are shown in Table 3. As both LoRA and prefix tuning have much fewer parameters to optimize (0.1% and 0.01% of the original number of parameters respectively for OPT1.3B) so that the number of convergence steps required will be lower, the performance gap between MeZO+PEFT and PseuZO+PEFT is much smaller than that of MeZO and PseuZO.

## 4.2 Memory usage and time computation

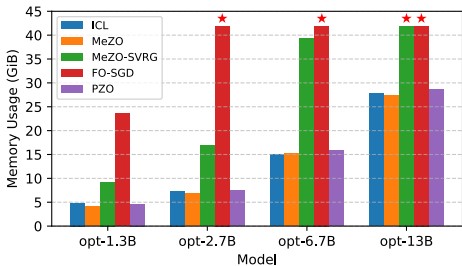

| $t_{\text{PseuZO}}$ | OPT1.3B | OPT2.7B | OPT6.7B |
|---|---|---|---|
| 10 | 9300 | 9600 | 9900 |
| 20 | 8700 | 9400 | 9700 |
| 50 | 6800 | 8400 | 9300 |
| 100 | 3600 | 7000 | 8700 |
| 150 | 400 | 5100 | 8000 |

Figure 2: Memory overhead for different models under various ZO algorithms and FO-SGD. ★ means out of memory.

Table 4: $t_{\text{PseuZO}}$ and its corresponding number of the total steps to guarantee the time that does not exceed 10K execution time for MeZO.

**PseuZO scales up to almost the same model size as MeZO can scale up to as PseuZO only introduces a small dimension-independent memory overhead.** Memory usage for different scale models is illustrated in Figure 2 which verifies our conclusion that compared to MeZO, excessive memory overhead of PseuZO is independent of the model size. Additionally, we also compare memory overhead for the different batch size and max length. As shown in Table 5, the memory requirement of PseuZO is more sensitive to the batch size and max length due to the sliding window. However, as model size has a greater impact on memory, PseuZO still needs less memory than MeZO-SVRG (MeZO-SVRG needs to store full parameters during training [19]).

**The computation time required for each epoch of PseuZO is approximately twice that of each epoch of MeZO for OPT-1.3B but the gap for large scale models will gradually decrease.** We fix the total steps of MeZO as **10K** and obtain the corresponding total steps of PseuZO for different

| | Memory Usage in GiB for OPT-1.3B | | | | | Memory Usage in GiB for OPT-6.7B | | | | |
| | Fixed context length (cl=128) | | | Fixed batch size (bs=16) | | Fixed context length (cl=128) | | | Fixed batch size (bs=16) | |
| Method | bs=16 | bs=32 | bs=64 | cl=256 | cl=512 | bs=16 | bs=32 | bs=64 | cl=256 | cl=512 |
|---|---|---|---|---|---|---|---|---|---|---|
| FO-SGD | 30.30 | ———————————————— OOM ———————————————— | | | | | | | | |
| MeZO | 5.35 | 7.58 | 11.87 | 9.06 | 17.03 | 17.09 | 20.35 | 27.51 | 21.28 | 27.01 |
| MeZO-SVRG | 10.55 | 12.76 | 17.55 | 12.46 | 19.67 | ————————— OOM ————————— | | | | |
| PseuZO | 7.61 | 10.58 | 17.85 | 11.69 | 19.36 | 18.08 | 23.22 | 33.52 | 25.16 | 32.74 |

Table 5: Memory usage (GiB) comparison on BoolQ for different ZO methods with OPT1.3B and OPT-6.7B showing that PseuZO can scale up to larger models as MeZO.

| Dataset | SPSA | $ZO_{sp}$ | PseuZO | PseuZO (w/LL) | BP |
|---|---|---|---|---|---|
| MNIST | $86.4 \pm 0.15$ | $87.8 \pm 0.09$ | $98.7 \pm 0.02$ | / | $98.5 \pm 0.02$ |
| CIFAR-10 | $41.3 \pm 0.74$ | $42.6 \pm 0.69$ | $82.5 \pm 0.15$ | $88.7 \pm 0.13$ | $89.9 \pm 0.06$ |
| CIFAR-100 | $5.39 \pm 0.69$ | $7.61 \pm 0.73$ | $61.4 \pm 0.14$ | $68.5 \pm 0.13$ | $71.9 \pm 0.09$ |

Table 7: Training from scratch on typical computer vision classification datasets for various feedback methods. We do not use local loss for MNIST as there are only two hidden layers.

$t_{\text{PseuZO}}$ under the same computation time where $t_{\text{PseuZO}}$ is the number of total epochs using PseuZO and results are shown in Table 4.

## 4.3 Ablation study

As introduced before, our sliding window has four key factors denoted as 1) $L$: sliding window length; 2) $t_{\text{PseuZO}}$: number of epochs to execute PseuZO for each cycle; 3) $R$: number of cycles; 4) $\lambda(t)$: the formula for $\lambda$ to descend with respect to epoch $t$. We perform ablation studies on SST2 to explore their individual impact on precision and the results are shown in Figure 3 and Table 6. If we keep $\lambda$ a small constant like $\lambda_{\min} = 0.1$, the accuracy is close to that of MeZO. In fact, when $\lambda \to 0$, PseuZO gradually degenerates to MeZO without momentum. More ablation experiments, including MeZO with momentum and PseuZO without momentum, can be found in Appendix A. We empirically found that performance is largely insensitive to the sliding window parameters. Since there is no need to set $L$, $t_{\text{PseuZO}}$ and $R$ too large, smaller values are chosen to balance low memory and time overhead.

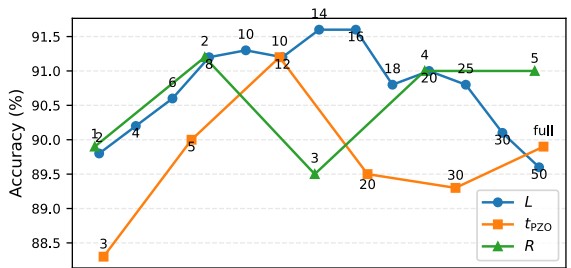

Figure 3: Ablation studies for $L$, $t_{\text{PseuZO}}$ and $R$. "full" means using PseuZO for all epochs.

| | $\lambda(t)$ | Accuracy(%) |
|---|---|---|
| Constant1 | $\lambda(t) = \lambda_{\max}$ | 83.9 |
| Constant2 | $\lambda(t) = \lambda_{\min}$ | 82.7 |
| Linear | $\lambda(t) = \lambda_{\max}(1 - \frac{t}{t_{\text{PseuZO}}})$ | 90.3 |
| Reciprocal | $\lambda(t) = \lambda_{\max}\frac{1}{1+0.5t}$ | 91.2 |

Table 6: Ablation study for different descent formula $\lambda(t)$ when $t < t_{\text{PseuZO}}$.

## 4.4 Training from scratch

During ZO optimization, the variance introduced by the large number of parameters is difficult to control and thus an appropriate prompt design is significant to guide generation [34, 19] for instruction fine-tuning. Furthermore, it remains a problem for ZO algorithms like SPSA to train from scratch even for small datasets. However, with a simple reformulation named Node Perturbation [32], PseuZO shows great performance on these tasks. The reformulation details and further explanation

can be found in Appendix B. As shown in Table 7, PseuZO in Node Perturbation manner outperforms SPSA and other signal feedback methods and with incorporation of local learning [26], PseuZO can even perform as well as BP. It demonstrates the potential of PseuZO to be used for more difficult but significant settings.

# 5 Conclusion

In this paper, we propose a new algorithm framework PesuZO and apply it to various instruction fine-tuning tasks for LLMs. According to our theoretical analyses, PseuZO finds an $\epsilon$-stationary point in $O\left(\max\left\{\alpha_1 L\epsilon^{-2}, \alpha_1 L\sigma_2^2\epsilon^{-4}\right\}\right)$ function evaluations. Experimental results demonstrate that PseuZO outperforms MeZO and MeZO-SVRG among various tasks. We propose sliding window technique, making the memory overhead independent of the model size. As a limitation, PseuZO needs to store the sliding window and the extra memory is sensitive to the batch size and max length. A possible fix to this problem is to add an low-dimension auxiliary layer as the new storing target. Though this method can further reduce memory overhead, it changes the model structure which might reduce its representation capability.

## Acknowledgments and Disclosure of Funding

Z. Lin was supported by National Key R&D Program of China (2022ZD0160300), the NSF China (No. 62276004) and the State Key Laboratory of General Artificial Intelligence.

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
