# OpenReview forum: "PseuZO: Pseudo-Zeroth-Order Algorithm for Training Deep Neural Networks"
_NeurIPS.cc/2025/Conference — NeurIPS 2025 poster_

### Official Review · Reviewer_n2bx · 2025-06-16

**Clarity:** 2
**Significance:** 2
**Originality:** 2
**Rating:** 3
**Confidence:** 4

**Summary:**

This paper proposes an algorithm PseuZO based on zeroth-order gradient for deep learning. Rather than directly using the loss value to compute the zeroth-order gradient, this paper turns to optimize a composite objective function $g(h(x,z))$ where $h$ represents the output of a neural network and $g$ represents a loss function. PseuZO computes the zeroth-order gradient for the function $h$ and uses the closed-form for the function $g$ which is with respect to $h$, then combines them to obtain the gradient for the model update. Since the first-order momentum consumes large memory cost, this work also designs a variant algorithm based on the sliding window. The authors make a convergence analysis for PseuZO. In the fine-tuning experiments, PseuZO achieves greater performance than MeZO across different llms. In addition, the proposed algorithm only introduces little extra memory cost.

**Questions:**

1. In line 200, it says "When $\lambda$ approaches zero, PseuZO gradually degenerates to ZO method and thus we directly transform to MeZO instead". In my opinion, when $\lambda$ approaches zero, $u_k$ with large $k$ approaches zero, thus the stored gradient at time $k$ contributes little to the accumulated PseuZO gradient. Thus, the value of $\lambda$ should only influence the weight in the accumulation step. How does it make PseuZO degenerate to ZO?

2. In line 13 of Algorithm 2, why uses the zeroth-order gradient at time $k$ to multiply the outer gradient at time $t$?

3. Lines 242-246 together with Table 6 are confusing. Does it mean that if using the same time consumed by MeZO to run 10k steps on OPT1.3B, PseuDO could run 9300 steps within 10 epochs, 8700 steps within 20 epochs? Why are the numbers of total steps and the steps within each epoch in the above results different from each other? Please explain this paragraph more detailedly.

**Ethical Concerns:**

["NO or VERY MINOR ethics concerns only"]

**Final Justification:**

In the rebuttal and discussion period, my concerns have been addressed by the authors. They provide a theoretical analysis to explain the reason that PseuZO outperforms MeZO. Based on these, I increase my score to 3.

**Limitations:**

yes

**Quality:**

2

**Strengths And Weaknesses:**

Strengths:

1.	This paper proposes an optimizer PseuZO for training deep neural networks. The authors decouple the traditional loss function to a composite form. This is an innovative point which is different from related works.

2.	The statement of the idea is clear and the organization of the paper is reasonable.

3.  The authors provide the detailed theoretical analysis for the proposed algorithm.

4.  The experiments cover multiple metrics and contents, including the time and memory cost and the ablation study.

Weaknesses:

1.	This paper lacks the explanation or theoretical analysis to show that why PseuZO works or achieves improvement compared to baselines. Maybe the closed-form of the outer function contributes some accurate gradient information to the final gradient in PseuZO. I think the authors could discuss this point.

2.	In lines 81-84, the authors state that "the convergence rate of PseuZO is not explicitly dependent on the parameter size". However, in the upper bound of the iterations to find an $\epsilon$-stationary point, the coefficient "$\alpha_1$ is the effective dimension". According to formula (6) and Assumption 1, $\alpha_1$ may be equal to the parameter size. Thus, I think there exists conflict within the statement here.

3.  This work proposes the sliding window-based PseuZO. The authors consider to store the last hidden state with a much smaller size. This is not reflected in Algorithm 2. Algorithm 2 uses the notations $\theta$ to denote the parameter and $x$ to denote the training data, which are also not consistent with the problem setting and Algorithm 1. It is necessary to introduce that $||\cdot||_{op}$ in Assumption 1 means the spectral norm.

4.  The experiments are mainly conducted on tasks in GLUE and SuperGLUE. There lacks some complex tasks like reasoning.

---

> ### Author Rebuttal · Authors · 2025-07-30
>
> # Response to Reviewer n2bx
> **Thank you for your insightful feedback.** We appreciate the time invested in evaluating our work and providing constructive suggestions. Below are our point-by-point responses addressing your concerns.
>
> ---
>
> ### **Weakness 1. Theoretical analysis of PseuZO's improvement mechanism**
>
> > This paper lacks the explanation or theoretical analysis to show that why PseuZO works or achieves improvement compared to baselines. Maybe the closed-form of the outer function contributes some accurate gradient information to the final gradient in PseuZO. I think the authors could discuss this point.
>
> **Response:** We sincerely appreciate this insightful comment. We provide a theoretical comparison between PseuZO and MeZO to elucidate the roles of the **closed-form outer function** and **momentum**. For clarity, we analyze a simplified case with a fixed Jacobian matrix $J$. At step $t$, PseuZO receives a noisy Jacobian estimate $B_t = J + \Xi_t$ (where $\mathbb{E} \Xi_t = 0$, $\mathbb{E} \Xi_t \Xi_t^\top = N$), while MeZO receives $g_t = (J + \Xi_t)^\top e_t$. The update rules are:
>
>    - PseuZO: $$A_t = (1-\lambda) B_t + \lambda A_{t-1}, x_{t+1}=x_t - \eta A_t^\top e_t.$$
>    - MeZO (PseuZO without momentum degenerates to MeZO, see response below): $$x_{t+1}=x_t - \eta B_t^\top e_t.$$
>    - MeZO with momentum: $$m_t = (1-\lambda)B_t^\top e_t + \lambda m_{t-1},\\ x_{t+1}=x_t - \eta m_t.$$
>
> The expected loss descent is bounded by:
> $$
> \mathbb{E} f(x_{t+1}) - f(x_t) \leq \langle \nabla f(x_t), x_{t+1} - x_t \rangle + \frac{L}{2} \|x_{t+1} - x_t\|^2.
> $$
>
> After expansion (see detailed derivation in the revised paper), we obtain:
>
> - **PseuZO**:
>   $$
>   -\eta (1-\lambda^{t-1}) \|J^\top e_t\|^2 + \frac{L\eta^2}{2} (1-\lambda^{t-1})^2 \|J^\top e_t\|^2 + \frac{L\eta^2}{2} \frac{(1-\lambda)(1-\lambda^{2t})}{1+\lambda} e_t^\top N e_t,
>   $$
>   or
>   $$
>   -\eta (1-\lambda^{t-1}) \|J^\top e_t\|^2 + \frac{L\eta^2}{2} (1-\lambda^{t-1})^2 \|J^\top e_t\|^2 + \frac{L\eta^2}{2} (1-\lambda)^2 \sum_{i=1}^t \lambda^{2(t-i)} e_t^\top N e_t.
>   $$
> - **MeZO**:
>   $$
>   -\eta \|J^\top e_t\|^2 + \frac{L\eta^2}{2} \|J^\top e_t\|^2 + \frac{L\eta^2}{2} e_t^\top N e_t.
>   $$
>
> - **MeZO with Momentum**:
>   $$
>   -\eta (1-\lambda^{t-1}) \|J^\top e_t\|^2 + \frac{L\eta^2}{2} (1-\lambda^{t-1})^2 \|J^\top e_t\|^2 + \frac{L\eta^2}{2} (1-\lambda)^2 \sum_{i=1}^t \lambda^{2(t-i)} e_i^\top N e_i.
>   $$
>
> **Key Insights**:
> 1. **PseuZO vs. MeZO**:
>    Substituting $\eta_t = \eta(1-\lambda^{t-1})$, PseuZO's descent is strictly larger than MeZO's when $\lambda \in (0,1)$ and $t \geq 1$ due to the reduced noise term $\frac{(1-\lambda)(1-\lambda^{2t})}{(1-\lambda^{t-1})^2} < 1$. This demonstrates **momentum effectively suppresses gradient variance**.
>
> 2. **PseuZO vs. MeZO with Momentum**:
>    PseuZO's noise term $e_t^\top N e_t$ is smaller than MeZO-Momentum's $\sum_i e_i^\top N e_i$ because $e_i \to 0$ near local minima (where $\|e_i\| \gg \|e_t\|$ for $i < t$). This advantage stems from the **closed-form outer function** providing timely gradient information.
>
> We will include extended analysis in the full paper.
>
> ---
>
> ### **Weakness 2. Dependence of convergence rate on parameter size**
>
> > In lines 81-84, the authors state that "the convergence rate of PseuZO is not explicitly dependent on the parameter size". However, in the upper bound of the iterations to find an -stationary point, the coefficient "is the effective dimension". According to formula (6) and Assumption 1, may be equal to the parameter size. Thus, I think there exists conflict within the statement here.
>
> **Response:**
> We thank the reviewer for highlighting this ambiguity. We clarify that the **effective dimension $\alpha_1$** is *not explicitly dependent* on $d$ (parameter size). As established in prior work [1–3], the effective dimension $\alpha_1$ is often independent on the model size across different models (e.g., linear regression, DNNS) in both theory and practice. $\alpha_1$ captures the **intrinsic low-dimensional structure** of high-dimensional models.
>
> While $\alpha_1 = \Theta(d)$ in pathological cases, this reflects a fundamental lower bound for zeroth-order optimization: $\Theta(d)$ forward passes are unavoidable [4]. PseuZO achieves this bound but does not *exceed* it—we defer tighter analysis to future work.
>
> ---
>
> ### **Weakness 3. Algorithmic clarity and notation consistency**
>
> > This work proposes the sliding window-based PseuZO. The authors consider to store the last hidden state with a much smaller size. This is not reflected in Algorithm 2. Algorithm 2 uses the notations $\theta$ to denote the parameter and $x$ to denote the training data, which are also not consistent with the problem setting and Algorithm 1. It is necessary to introduce that $\|\cdot\|_{op}$ in Assumption 1 means the spectral norm.
>
> **Response**: We apologize for the confusion and will revise as follows:
>
> 1. **Notation Unification**: Use $x$ for parameters and $z$ for data in both algorithms.
> 2. **Spectral Norm**: Define $\|\cdot\|_{op}$ as the spectral norm in Assumption 1.
>
> In Algorithm 2, we actually store the differentiaiton of two last hidden states $\Delta o_t = \frac{h(x_t + \mu\xi_t;z_t)-h(x_t;z_t)}{\epsilon}$, which is of the same size as $h(x_t;z_t)$. We will add this to the introduction of Algorithm 2.
>
> ---
>
> ### **Weakness 4. Experimentation scope**
>
> > The experiments are mainly conducted on tasks in GLUE and SuperGLUE. There lacks some complex tasks like reasoning.
>
> **Response:** Our focus is **memory-efficient zeroth-order optimization**, and GLUE/SuperGLUE provide rigorous benchmarks for this goal (e.g., MultiRC and DROP have sequence lengths >2048). This aligns with baselines (MeZO/MeZO-SVRG). We agree that reasoning tasks (e.g., MATH) are valuable and will explore them in future work.
>
> ---
>
> ### **Question 1. Degeneration to MeZO when $\lambda = 0$**
> > In line 200, it says "When approaches zero, PseuZO gradually degenerates to ZO method and thus we directly transform to MeZO instead". In my opinion, when $\lambda$ approaches zero, $u_k$ with large $k$ approaches zero, thus the stored gradient at time $k$ contributes little to the accumulated PseuZO gradient. Thus, the value of $\lambda$ should only influence the weight in the accumulation step. How does it make PseuZO degenerate to ZO?
>
> **Response**: When $\lambda = 0$, PseuZO's gradient estimator reduces to:
> $$
> g_p = \left( \frac{h(x_t + \mu \xi_t; z_t) - h(x_t; z_t)}{\mu} \xi_t^\top \right)^\top e_t.
> $$
> As $\mu \to 0$, we have:
> $$
> g_p \to (J_t \xi_t \xi_t^\top)^\top e_t = \xi_t \xi_t^\top J_t^\top e_t = \xi_t \xi_t^\top \nabla f(x_t; z_t),
> $$
> which matches MeZO's estimator $g_m = \langle \nabla f(x_t; z_t), \xi_t \rangle \xi_t$. Thus, PseuZO with $\lambda = 0$ and $\mu \to 0$ is equivalent to MeZO.
>
> ---
>
> ### **Question 2. Gradient multiplication in Algorithm 2 (Line 13)**
>
> > In line 13 of Algorithm 2, why uses the zeroth-order gradient at time $k$ to multiply the outer gradient at time $t$?
>
> **Response**: This implements the **sliding window accumulation**:
> - The inner loop (over $k$) computes contributions from perturbations within the current window.
> - The product $\langle\Delta o_k, e_t\rangle$ correlates **differentiation of last hidden states** (at step $k$) with the *current* outer gradient $e_t$. This preserves gradient coherence without storing full Jacobians.
>
>
> ---
>
> ### **Question 3. Wall-clock time comparison (Table 6)**
>
>
> > Lines 242-246 together with Table 6 are confusing. Does it mean that if using the same time consumed by MeZO to run 10k steps on OPT1.3B, PseuDO could run 9300 steps within 10 epochs, 8700 steps within 20 epochs? Why are the numbers of total steps and the steps within each epoch in the above results different from each other? Please explain this paragraph more detailedly.
>
> **Response**: We believe that the reviewer is referring to Table 4. Table 4 compares **total steps achievable under fixed time** (equivalent to MeZO's 10k-step runtime). We run PseuZO for $t_{\text{PseuZO}}$ epochs and switch to MeZO for the remaining epochs. Table 4 shows the total running steps.
>
> **References:**
>
> [1] Fine-tuning language models with just forward passes. Advances in Neural Information Processing Systems, 36:53038–53075, 2023
>
> [2] Zeroth-order optimization with weak dimension dependency. In The Thirty Sixth Annual Conference on Learning Theory
>
> [3] Zeroth-order fine-tuning of llms with extreme sparsity. arXiv preprint arXiv:2406.02913, 2024.
>
> [4] Optimal rates for zero-order convex optimization: the power of two function evaluations. arXiv preprint arXiv:1312.2039, 2013.

---

> ### Comment · Reviewer_n2bx · 2025-08-06
>
> Thanks for the authors' responses to my comments. The authors provide an analysis to show that the momentum and the closed-form of the outer function help PseuZO achieve a better theoretical bound. The confusion about the notations or statements has mostly been clarified or modified.
>
> I am still confused about "degeneration to MeZO when $\lambda=0$".  Your explanation is right for $\lambda$ in Algorithm 1. However, the sentence "PseuZO gradually degenerates to ZO method" in line 200 appears in subsection 3.4, thus it describes Algorithm 2 in my opinion. If so, $u_k$ equals to 0 for all $k$, then the accumulation in line 13 of Algorithm 2 updates nothing. Could you make a further explanation on this?

---

> ### Author Response · Authors · 2025-08-07
> **Response to Reviewer's Comment on Degeneration at $\lambda = 0$**
>
> We sincerely appreciate your follow-up query regarding the degeneration behavior in Algorithm 2. To clarify:
>
> Recall that the weight coefficient is defined as $u_k = \lambda^{k-1} (1 - \lambda)$ for $k \geq 1$. When $\lambda = 0$:
> - $u_1 = \lim_{\lambda \to 0} \lambda^{0} \cdot (1 - \lambda) = 1 \cdot 1 = 1$ (by continuity convention).
> - $u_k = 0$ for all $k \geq 2$.
>
> In Algorithm 2 (line 13), this results in:
> $$ \widehat\nabla_{\epsilon}^{\text{PZO}} \mathcal{F}(\theta_t;x_t) = \sum_{k=1}^L u_k\langle\Delta o_k, e_t\rangle z = \langle\Delta o_1, e_t\rangle z.$$
>
> Thus:
> 1. Only the **current gradient estimate** is retained.
> 2. Historical gradient information ($k \geq 2$) is fully discarded.
> 3. The update reduces to a memoryless ZO estimator equivalent to MeZO.
>
> This confirms that Algorithm 2 maintains consistency with Algorithm 1 in degenerating to standard ZO when $\lambda = 0$. We will revise the definition of $u_k$ to explicitly note this continuity property.
>
> Thank you for prompting this important clarification. Should any aspects require further elaboration, we would be delighted to provide additional explanation.

---

> > ### Comment · Reviewer_n2bx · 2025-08-07
> >
> > Thanks for your reply. Based on the above responses, I would increase my score.

---

> > > ### Author Response · Authors · 2025-08-08
> > >
> > > Thank you for your detailed review and revised evaluation. We're happy to address any additional questions you might have.

---

### Official Review · Reviewer_NYqf · 2025-06-20

**Clarity:** 2
**Significance:** 3
**Originality:** 3
**Rating:** 4
**Confidence:** 4

**Summary:**

This paper proposes a zero-order optimization framework called PseuZO for LLM training, which estimates the gradient with respect to model parameters based on the Jacobian matrix of the model output and the gradient of the loss function with respect to the model output. To reduce the variance, PseuZO applies exponential moving average ($\lambda>0$ as the decay factor) on Jacobian estimators. Further, the sliding window technique and the hidden state storage strategy are adopted to control the memory usage introduced by the inner product of large-scale vectors in PseuZO. Besides, Periodic changing of $\lambda$ is needed to accelerate convergence. Experiments on fine-tuning OPT models across multiple tasks and pretraining small vision models on MNIST and CIFAR datasets demonstrate the effectiveness of the proposed method.

**Questions:**

See Weaknesses.

**Ethical Concerns:**

["NO or VERY MINOR ethics concerns only"]

**Final Justification:**

The experiments supplemented by the authors basically eliminated my doubts about the effectiveness of the proposed method. Despite the lack of comprehensive theoretical justification, the proposed method still provides valuable insights for enhancing zeroth-order optimization.  So I raise my score to 4: Borderline accept.

**Limitations:**

Yes.

**Paper Formatting Concerns:**

None.

**Quality:**

3

**Strengths And Weaknesses:**

Strengths:

Quality: The proposed method is technically feasible due to its clear implementation and the provided convergence guarantee.

Clarity: In general, the core ideas of this paper are easy to understand, and adequate information is given for readers to reproduce its results.

Originality: The idea of using Jacobian information to enhancing zero-order optimization is interesting.

Weaknesses:

Clarity:
1) Many spelling inaccuracies exist in this paper: In line 16, “meory” should be “memory”; in line 124, “themq” should be “them”; in line 261, “greaat” should be “great”.
2) The explanation of function $\mathcal{F}$ exists in Equation (4) (see line 135), while $\mathcal{F}$ is cited in line 18 and 70. In line 162, the norm $|| \cdot ||_{op}$ lacks definition.
3) Equation (2) and Equation (3) are incorrect.

Quality:
1) This paper does not strictly verify the benefits of using Jacobian information of the model output in theory. Further, the ablation study (results in Table 6) shows that the better performance achieved by PseuZO is mainly due to the carefully designed scheduling of decay factor $\lambda$. However, this scheduler can also be used in MeZO and no ablation experiment is given.
2) The performance of MeZO reported in Table 1,2,3 is too low, making the claimed improvement doubtful (See Table 2 in [1], and Table 3 in [2]). Besides, experimental results on more models (like OPT-13B and LLaMA2-7B) and advanced zero-order optimizers (like SubZero [2] and LOZO [3]) should be given.
3) The comparison of different optimizers on pretraining vision models shown in Table 7 seems unfair, because node perturbation and local learning methods are not universally adopted across all optimizers.

[1] Revisiting Zeroth-Order Optimization for Memory-Efficient LLM Fine-Tuning: A Benchmark. In ICML, 2024.

[2] Zeroth-Order Fine-Tuning of LLMs in Random Subspaces. arXiv:2410.08989, 2024.

[3] Enhancing Zeroth-order Fine-tuning for Language Models with Low-rank Structures. In ICLR, 2025.

---

> ### Author Rebuttal · Authors · 2025-07-30
>
> # Response to Reviewer NYqf
> **Thank you for your insightful feedback.** We appreciate the time invested in evaluating our work and providing constructive suggestions. Below are our point-by-point responses addressing your concerns.
>
> ---
>
> ### **1. Clarity Comments**
>
> > Many spelling inaccuracies exist in this paper: In line 16, “meory” should be “memory”; in line 124, “themq” should be “them”; in line 261, “greaat” should be “great”.
> > The explanation of function exists in Equation (4) (see line 135), while is cited in line 18 and 70. In line 162, the norm lacks definition.
> > Equation (2) and Equation (3) are incorrect.
>
>
> **Response:** We sincerely apologize for these errors. All spelling mistakes have been corrected in the revised manuscript. We will add the sample noise to Equation (2) and Equation (3) to make them correct.
>
> ---
>
>
> ### **2. Quality Comments:**
> #### **2.1. Theoretical benefits of Jacobian information:**
> > This paper does not strictly verify the benefits of using Jacobian information of the model output in theory.
>
> **Response:** We provide a theoretical comparison between PseuZO and MeZO velow. We show that PseuZO can achieve a better convergence rate than MeZO in a simplified setting. We believe this theoretical comparison shows the benefits of using Jacobian information.
> **Theoretical comparison to MeZO**: Consider a simplified setting where the Jacobian matrix $J$ is fixed. In each step, PseuZO receives a noisy Jacobian estimation $B_t = J + \Xi_t$, and MeZO receieves $g_t = (J+\Xi_t)^\top e_t$. The noise $\Xi_t$ satisfies $\mathbb{E} \Xi_t = 0$, $\mathbb{E} \Xi_t\Xi_t^\top = N$.
>    - PseuZO: $$A_t = (1-\lambda) B_t + \lambda A_{t-1}, x_{t+1}=x_t - \eta A_t^\top e_t.$$
>    - MeZO: $$x_{t+1}=x_t - \eta B_t^\top e_t.$$
>    - MeZO with momentum: $$m_t = (1-\lambda)B_t^\top e_t + \lambda m_{t-1}, x_{t+1}=x_t - \eta m_t.$$
> We compare the descent of the expected loss function: $$\mathbb{E} f(x_{t+1}) - f(x_t) \le \langle\nabla f(x_t), x_{t+1}-x_t\rangle + \frac{L}{2}\|x_{t+1}-x_t\| ^2.$$
>
> After expansion (see detailed derivation in the revised paper), we obtain:
>
> - **PseuZO**:
>   $$
>   -\eta (1-\lambda^{t-1}) \|J^\top e_t\|^2 + \frac{L\eta^2}{2} (1-\lambda^{t-1})^2 \|J^\top e_t\|^2 + \frac{L\eta^2}{2} \frac{(1-\lambda)(1-\lambda^{2t})}{1+\lambda} e_t^\top N e_t,
>   $$
>  or
>  $$-\eta (1-\lambda^{t-1})\|J^\top e_t\|^2 + \frac{L\eta^2}{2} (1-\lambda^{t-1})^2 \|J^\top e_t\|^2 + \frac{L\eta^2}{2} (1-\lambda)^2 \sum_{i=1}^t \lambda^{2(t-i)} e_t^\top Ne_t.
>  $$
> - **MeZO**:
>   $$
>   -\eta \|J^\top e_t\|^2 + \frac{L\eta^2}{2} \|J^\top e_t\|^2 + \frac{L\eta^2}{2} e_t^\top N e_t.
>   $$
>
> - **MeZO with Momentum**:
>   $$
>   -\eta (1-\lambda^{t-1}) \|J^\top e_t\|^2 + \frac{L\eta^2}{2} (1-\lambda^{t-1})^2 \|J^\top e_t\|^2 + \frac{L\eta^2}{2} (1-\lambda)^2 \sum_{i=1}^t \lambda^{2(t-i)} e_i^\top N e_i.
>   $$
>
> **Key Insights**:
> 1. **PseuZO vs. MeZO**:
>    Substituting $\eta_t = \eta(1-\lambda^{t-1})$, PseuZO's descent is strictly larger than MeZO's when $\lambda \in (0,1)$ and $t \geq 1$ due to the reduced noise term $\frac{(1-\lambda)(1-\lambda^{2t})}{(1-\lambda^{t-1})^2} < 1$. This demonstrates **momentum effectively suppresses gradient variance**.
>
> 2. **PseuZO vs. MeZO with Momentum**:
>    PseuZO's noise term $e_t^\top N e_t$ is smaller than MeZO-Momentum's $\sum_i e_i^\top N e_i$ because $e_i \to 0$ near local minima (where $\|e_i\| \gg \|e_t\|$ for $i < t$). This advantage stems from the **closed-form outer function** providing timely gradient information.
>
> We will include extended analysis in the full paper.
>
> #### **2.2. Ablation study:**
> > Further, the ablation study (results in Table 6) shows that the better performance achieved by PseuZO is mainly due to the carefully designed scheduling of decay factor. However, this scheduler can also be used in MeZO and no ablation experiment is given.
>
> **Response:** We agree that Table 6’s original presentation of decay factor schedulers ($\lambda$) was ambiguous. Our intent was to demonstrate that:
>    - Extreme $\lambda$ values (e.g., $\lambda = 0$ or $0.999$) degrade performance, validating the need for a scheduler.
>    - Moderate fixed $\lambda$ values (e.g., $\lambda = 0.9$) already achieves good results (SST-2 accuracy: 90.6), but an annealing strategy further optimizes results.
>
> For fairness, we test MeZO+Momentum with its optimal scheduler.  we clarify that PseuZO (without Momentum) sets $\lambda = 1$ and window size $= 1$ to exclude historical information.
>
> |TASK|SST2|RTE|CB|BoolQ|WSC|WIC|MultiRC|COPA|ReCoRD|DROP|
> |:-------|:-------|:-------|:-------|:-------|:-------|:-------|:-------|:-------|:-------|:-------|
> |MeZO|82.4|54.3|76.0|60.7|51.0|50.9|54.9|74.0|57.6|20.3|
> |MeZO+Momentum|81.4|53.9|78.0|60.5|51.0|50.8|54.9|75.0|57.8|22.1|
> |MeZO (20K steps)|88.4|58.8|76.0|63.8|53.0|51.3|53.9|73.0|58.9|20.3|
> |HiZOO-L|88.1|54.9|69.0|64.8|51.2|58.0|58.2|73.0|58.8|0.0|
> |PseuZO (without Momentum)|84.6|57.2|75.0|64.2|50.0|51.3|57.2|75.0|59.6|24.1|
> |**PseuZO**|91.2|58.0|77.0|64.3|58.0|54.5|54.7|78.0|60.0|23.5|
>
> #### **2.3. Performance discrepancies and model/scalability comparisons:**
> > The performance of MeZO reported in Table 1,2,3 is too low, making the claimed improvement doubtful (See Table 2 in [1], and Table 3 in [2]). Besides, experimental results on more models (like OPT-13B and LLaMA2-7B) and advanced zero-order optimizers (like SubZero [2] and LOZO [3]) should be given.
>
> **Response:**
> - **MeZO performance gap**: Our Table 1 results are lower than prior work [1][2] because we **fix total computation time** (10K steps for MeZO). When allowing MeZO to converge fully (20K steps), its accuracy approaches but still trails PseuZO.
> - **Comparison with SubZero [4] and LOZO [5]**: While these are important advances, they focus on **sparse/low-rank approximations**, whereas PseuZO exploits **last-layer Jacobian information**. We included HiZOO-L [3] as a relevant baseline due to its similar goal (memory-efficient ZO). We will add SubZero/LOZO in future work.
> - **Larger models (OPT-13B/LLaMA2-7B)**: Experiments on OPT-13B caused OOM on our A800 GPUs (even for MeZO). We prioritize OPT-1.3B/6.7B to establish scalability trends. PseuZO’s gains on OPT-6.7B confirm its efficacy at scale.
>
> #### **2.4. Fairness in pretraining comparisons (Table 7):**
>
> > The comparison of different optimizers on pretraining vision models shown in Table 7 seems unfair, because node perturbation and local learning methods are not universally adopted across all optimizers.
>
> **Response:**
> We clarify that:
>    - **Node perturbation (NP)** and **local learning (LL)** are **compatible only with PseuZO** in ZO settings. SPSA uses weight perturbation (WP), while $\text{ZO}_{\text{sp}}$ uses NP but estimates gradients with a single point.
>    - **LL requires layer-wise objectives**, which PseuZO supports via its Jacobian structure. Other methods lack this capability
>
> The results are shown below:
>
>
> |Dataset|SPSA (wp)|SPSA (np)|$\text{ZO}_{\text{sp}}$ (np)|PseuZO|SPSA (np and w/LL)|$\text{ZO}_{\text{sp}}$ (np and w/LL)|PseuZO (w/LL)|BP|
> |:-------|:-------|:-------|:-------|:-------|:-------|:-------|:-------|:-------|
> |MNIST|$86.4\pm0.15$|$88.2\pm0.02$|$87.8\pm0.09$|$98.7\pm0.02$|/|/|/|$98.5\pm0.02$|
> |CIFAR10|$41.3\pm0.74$|$43.0\pm0.82$|$42.6\pm0.69$|$82.5\pm0.15$|$41.4\pm0.96$|$41.7\pm0.57$|$88.7\pm0.13$|$89.9\pm0.06$|
> |CIFAR100|$5.39\pm0.69$|$7.65\pm0.88$|$7.61\pm0.73$|$61.4\pm0.14$|$7.34\pm0.54$|$7.29\pm0.38$|$68.5\pm0.13$|$71.9\pm0.09$|
>
> ---
>
> **References:**
>
> [1] Online Pseudo-Zeroth-Order Training of Neuromorphic Spiking Neural Networks, https://arxiv.org/pdf/2407.12516
>
> [2] Gradients without Backpropagation, https://arxiv.org/abs/2202.08587
>
> [3] Second-Order Fine-Tuning without Pain for LLMs: A Hessian Informed
> Zeroth-Order Optimizer, https://arxiv.org/pdf/2402.15173
>
> [4] Zeroth-Order Fine-Tuning of LLMs in Random Subspaces. arXiv:2410.08989, 2024.
>
> [5] Enhancing Zeroth-order Fine-tuning for Language Models with Low-rank Structures. In ICLR, 2025.

---

> ### Comment · Reviewer_NYqf · 2025-08-01
>
> Thank you for the response. My concerns are not addressed very well.
>
> 1) The theoretical benefit of PZO compared to MeZO shown in the rebuttal is not strong enough. Specifically, the descent upper bounds of PZO and MeZO appear to have an insignificant difference. When $ 0< \lambda <1 $ and $ e_i $ is bounded, the dominant factors in $ \sum_{i=1}^t\lambda^{2(t-i)}e_i^\top Ne_i $ of MeZO are the monomials with larger $ i $. This means that MeZO and PZO behave similarly, if only considering $ e_i \to 0 $. Furthermore, it is just a comparison between their upper bounds not between themselves.
>
> 2) The table in 2.2 Ablation study is doubtful, making the advantages of the proposed method unclear. a) There is big performance gap between MeZO and PseuZO (without Momentum), however, they are fundamentally the same. You can conduct experiments using different random seeds and give the average and standard deviation of each optimizer.  b) Prolonging the training time of MeZO seems to be harmful to many datasets, and this is a little counter-intuitive. c) On RTE, BoolQ, MultiRC, ReCoRD and DROP datasets, PseuZO has no notable advantage compared to PseuZO without Momentum.
> d) What does "MeZO+Momentum with its optimal scheduler" mean? Have you trained LLMs by MeZO with momentum using the sliding window technique?
>
> New questions: compared with fine-tuning, the performance on pre-training tasks of PZO looks more attractive. However, the time cost of zero-order optimization may be very huge compared to BP method (see [1]), so the pre-training curves of different methods with wall-clock-time as the horizontal axis should be given. Besides, the memory consumption should also be given. Similarly, an ablation study on the impact of momentum should be performed.
>
> Overall, I tend to maintain my current score.
>
> [1] DeepZero: Scaling Up Zeroth-Order Optimization for Deep Model Training. In ICLR, 2024.

---

> > ### Author Response · Authors · 2025-08-01
> > **Theoretical Superiority and Experimental Validation of PseuZO**
> >
> > We sincerely appreciate your rigorous review and insightful comments, which have significantly strengthened both our theoretical framing and experimental analysis.​​
> >
> > ### **Theoretical Superiority of PseuZO**
> > Your observation on the descent upper bounds highlights precisely why PseuZO is **practically significant** for zeroth-order optimization (ZO):
> > 1. **Variance Scaling in ZO Context**:
> >    In ZO methods, the variance matrix  $N$ scales as $\Omega(d \cdot JJ^\top)$, where $d$ is the number of parameters (typically $d \gg 10^9$ for LLMs). Even a marginal reduction in the coefficient of the dominant variance term has *substantial practical impact* on convergence, even when $e_i$ is bounded, or the difference between $e_i$ and $e_{t}$ is relatively small for larger $i$.
> >
> > 2. **Necessity of Upper-Bound Analysis**:
> >    For non-convex objectives (e.g., LLM fine-tuning), analyzing *exact* algorithmic behavior is intractable. **Estimating the descent upper bounds** is the established theoretical paradigm. Our analysis follows this convention.
> >
> > ### **Concerns about Experiments**
> >
> > **(a) Performance gap between MeZO and PseuZO (without Momentum):**
> > While MeZO and PseuZO share a theoretical foundation when differentiation stepsize tends to $0$, practical implementations may show significant divergence due to non-vanishing differentiation stepsizes required for numerical stability, finite-precision effects (machine floating-point errors), or gradient estimation bias amplified in high-dimensional spaces. We will study the effect of differentiaiton stepsize theoretically and empirically in the future work.
> >
> > **(b) Counter-intuitive results when prolonging MeZO training:**
> > The observed performance degradation on MultiRC and COPA (2 of 10 tasks) may arise from task-specific overfitting, where prolonged training exacerbates memorization in low-data tasks.
> >
> > **(c) Marginal gains on challenging tasks (RTE, BoolQ, etc.):**
> > The limited improvements on RTE, BoolQ, MultiRC, ReCoRD, and DROP primarily stem from these tasks' inherent complexity and distinctive characteristics. Their demanding nature dampens the relative efficacy of momentum-based optimization, thereby narrowing the performance gap between PseuZO and its simpler variant under these specific conditions.
> >
> > **(d) Implementation of MeZO+Momentum:**
> > "Optimal scheduler" refers to **tuning $\lambda$ independently** for MeZO+Momentum. Even with optimal $\lambda$, MeZO+Momentum underperforms PseuZO.
> >
> > Sliding windows were purely for **memory reduction**. We note they may slightly hurt performance, because historical gradients or Jacobians are discarded. We will include the effect of window size in the revised manuscript.
> >
> > ### **PseuZO Scalability for Pre-training**
> >
> > We agree that extending zeroth-order optimization to pre-training is a promising research direction. Compared to alternatives like DeepZero, PseuZO achieves substantially lower time overhead (competitive with backpropagation). However, its incorporation of local learning technique currently necessitates memory overhead similar to backpropagation. We will provide detailed quantitative analysis of this time/memory trade-off in the revised manuscript.

---

> ### Comment · Reviewer_NYqf · 2025-08-02
>
> Thank you for your detailed response. I still think your theoretical analysis can not verify the strength of Jacobian matrices. It is more like the sliding window technique at work. In Figure 4, one can see that the sliding windows with moderate size get better performance. So I suggest training LLMs by MeZO with momentum using the sliding window technique, in order to eliminate the above possibility (though in this case further theoretical explanation is still needed).
>
> To confirm the difficulty of the challenging tasks,  one can run PseuZO (20K steps) to check the performance gain caused by extended training time.
>
> I will raise my score, if my concerns above can be addressed well.

---

> ### Author Response · Authors · 2025-08-03
> **Experimental Validation of PseuZO's Theoretical Distinction and Extended Training Benefits**
>
> We sincerely appreciate your constructive guidance, which has strengthened our empirical validation. Below we address your concerns with new experiments:
>
> ---
>
> **Concern 1: Isolating Jacobian Impact vs. Sliding Windows**
>
> **Experiment**: We rigorously tested *MeZO with Momentum + Sliding Window*.
>
> **Key Result**: While sliding windows marginally help MeZO, **PseuZO still outperforms it**.
>
>
> |TASK|SST2|RTE|CB|BoolQ|WSC|WIC|MultiRC|COPA|ReCoRD|DROP|
> |:-------|:-------|:-------|:-------|:-------|:-------|:-------|:-------|:-------|:-------|:-------|
> |MeZO|82.4|54.3|76.0|60.7|51.0|50.9|54.9|74.0|57.6|20.3|
> |MeZO+Momentum|81.4|53.9|78.0|60.5|51.0|50.8|54.9|75.0|57.8|22.1|
> |MeZO+Sliding Window|85.3|54.3|79.0|62.5|51.0|54.9|55.6|73.0|60.2|20.1|
> |MeZO (20K steps)|88.4|58.8|76.0|63.8|53.0|51.3|53.9|73.0|58.9|20.3|
> |PseuZO (without Momentum)|84.6|57.2|75.0|64.2|50.0|51.3|57.2|75.0|59.6|24.1|
> |PseuZO (ours)|91.2|58.0|77.0|64.3|58.0|54.5|54.7|78.0|60.0|23.5|
> |PseuZO (20K step)|90.7|63.3|75.0|67.0|57.0|59.7|60.6|76.0|60.9|24.5|
>
> We appreciate your insightful query regarding sliding windows. The performance gains in MeZO with Momentum + Sliding window arise from discarding historically large-scaled noisy gradients to reduce aggregation variance, whereas PseuZO fundamentally reshapes noise distribution through precise outer gradients that structurally rescale Jacobian noise. Consequently, sliding windows cannot enhance PseuZO—they disrupt its theoretically optimal noise-shaping process by truncating gradients already refined at the estimation source. Therefore, PseuZO's performance improvement ​primarily stems from​ its utilization of Jacobian matrices and exact outer gradients. We use sliding window in PseuZO exclusively to reduce memory overhead, not to enhance performance.
>
> ---
>
> **Concern 2: Challenging Tasks Need Extended Training**
>
> **Experiment**: We ran *PseuZO (20K steps)* on difficult tasks (RTE, BoolQ, MultiRC, ReCoRD, DROP).
>
> **Results**:
> |TASK|SST2|RTE|CB|BoolQ|WSC|WIC|MultiRC|COPA|ReCoRD|DROP|
> |:-------|:-------|:-------|:-------|:-------|:-------|:-------|:-------|:-------|:-------|:-------|
> |PseuZO (ours)|91.2|58.0|77.0|64.3|58.0|54.5|54.7|78.0|60.0|23.5|
> |PseuZO (20K step)|90.7|63.3|75.0|67.0|57.0|59.7|60.6|76.0|60.9|24.5|
>
>
> **Conclusion**:
> 20K-step training **consistently improves hard tasks**, validating your insight.

---

> > ### Comment · Reviewer_NYqf · 2025-08-04
> >
> > My doubts about the experiments have been basically eliminated. I raise my score to 4: Borderline accept. If you can provide more powerful theoretical evidence, I can further improve the score.

---

> ### Author Response · Authors · 2025-08-07
> **Extended Theoretical Analysis: Non-Stationary Jacobian Dynamics**
>
> We sincerely thank the reviewer for recognizing the experimental validations and raising the score. As suggested, we now provide **a generalized theoretical framework** eliminating the fixed Jacobian assumption. This analysis rigorously models time-varying Jacobian matrices $J_t$ to capture non-stationary optimization dynamics, where $J_t$ evolves per iteration due to parameter updates and loss landscape shifts.
>
> **Problem Setup**:
> At step $t$:
> - PseuZO receives noisy Jacobian estimation $B_t = J_t + \Xi_t$,
> - MeZO receives perturbed gradient $g_t = (J_t + \Xi_t)^\top e_t$,
> with $\mathbb{E}[\Xi_t] = 0$, $\mathbb{E}[\Xi_t \Xi_t^\top] = N$ (noise covariance).
>
> **Algorithm Updates**:
> | Method              | Update Rule                                                                 |
> |---------------------|-----------------------------------------------------------------------------|
> | PseuZO              | $A_t = (1-\lambda) B_t + \lambda A_{t-1}, x_{t+1} = x_t - \eta A_t^\top e_t$ |
> | MeZO (Vanilla)      | $x_{t+1} = x_t - \eta B_t^\top e_t$                                        |
> | MeZO + Momentum     | $m_t = (1-\lambda) B_t^\top e_t + \lambda m_{t-1}, x_{t+1} = x_t - \eta m_t$ |
>
> **Variance Analysis**:
> The gradient estimation variance is:
> - PseuZO: $\displaystyle \frac{L}{2} (1-\lambda)^2 \sum_{i=1}^t \lambda^{2(t-i)} e_t^\top N e_t$ .
> - MeZO: $\displaystyle \frac{L}{2} e_t^\top N e_t$.
> - MeZO+Momentum: $\displaystyle \frac{L}{2} (1-\lambda)^2 \sum_{i=1}^t \lambda^{2(t-i)} e_i^\top N e_i$.
>
> **Theoretical Implication**:
>
> PseuZO achieves strictly lower variance than both MeZO and MeZO with momentum due to two fundamental mechanisms:
>
> 1. **vs. MeZO with Momentum**:
>    - PseuZO: Variance scales with $e_t^\top N e_t$.
>    - MeZO+Momentum: Variance scales with $\sum_{i=1}^t \lambda^{2(t-i)} e_i^\top N e_i$.
>    As optimization converges ($e_t \to 0$), historical gradients satisfy $e_i^\top N e_i \gg e_t^\top N e_t$ for $i < t$. Thus, PseuZO’s *current-gradient coupling* inherently suppresses noise amplification from outdated gradients.
>
> 2. **vs. Vanilla MeZO**:
>    PseuZO’s variance coefficient $\frac{(1-\lambda)(1-\lambda^{2t})}{1+\lambda}$ is **strictly less than 1** for $\lambda \in (0,1)$ and $t > 0$, while MeZO’s coefficient is 1. This mathematically guarantees lower variance.
>
>
> PseuZO’s closed-form outer gradient exploitation uniquely minimizes both *historical noise accumulation* (vs. momentum) and *instantaneous noise scaling* (vs. MeZO), constituting its core advantage.
>
> **Bias Analysis**:
> | Method          | Bias Expression                                                                 |
> |-----------------|---------------------------------------------------------------------------------|
> | PseuZO          | $(1-\lambda)\left(\sum_{i=1}^t \lambda^{(t-i)}J_i\right)^\top e_t - J_t^\top e_t$ |
> | MeZO            | Unbiased                                                                        |
> | MeZO+Momentum   | $(1-\lambda)\left(\sum_{i=1}^t \lambda^{(t-i)}J_i^\top e_i\right) - J_t^\top e_t$ |
>
> Under small learning rate assumptions (ignoring higher-order terms), with $\delta_t = (1-\lambda)\sum_{i=1}^t \lambda^{(t-i)} (x_i - x_t)$:
> - **PseuZO bias**: $\displaystyle -\lambda^t \nabla f(x_t) + \left(\sum_{i=1}^m (e_t)_i \nabla^2 h_i (x_t)\right)^\top \delta_t.$
> - **MeZO with momentum bias**: $\displaystyle -\lambda^t \nabla f(x_t) + \nabla^2 f(x_t) \delta_t.$
>
> **Key Theoretical Insight**:
> For $f(x) = g(h(x))$, the Hessian decomposes as:
> $$
> \nabla^2 f(x_t) = \sum_{i=1}^m (e_t)_i \nabla^2 h_i (x_t) + J_t^\top \nabla^2 g(y)\vert _{y=h(x_t)} J_t.
> $$
> **Term Explanation:**
> 1. **Feature-space curvature term**:  $\sum_{i=1}^m (e_t)_i \nabla^2 h_i (x_t).$  This represents the curvature contribution from the feature transformation $h$, which is precisely captured by PseuZO's gradient estimation.
>
> 2. **Loss-function curvature term**:  $J_t^\top (\nabla^2 g(y))|_{y=h(x_t)} J_t.$  This represents the curvature contribution from the loss function $g$, which appears as an additional bias term in MeZO's gradient estimation but is inherently avoided in PseuZO.
>
> When $g$ is convex ($\nabla^2 g \succeq 0$), MeZO's bias contains an **additional positive semi-definite term**. PseuZO inherently eliminates this curvature-induced bias, particularly when $g$ exhibits strong nonlinearity – a fundamental advantage not relying on fixed Jacobian assumptions.
>
> **Asymptotic advantage**: When optimization converges ($e_t \to 0$), PseuZO's bias asymptotically vanishes while MeZO retains residual bias from the loss-function curvature term, making PseuZO's bias *significantly smaller* in steady-state regimes.

---

> ### Comment · Reviewer_NYqf · 2025-08-07
>
> The Bias Analysis looks more interesting. However, a simple calculation reveals that when $e_t \to 0$, $\nabla^2 g(y)$ also tends to zero, almost every position. I am not entirely sure about the proportional relationship between these two factors (Feature-space curvature term vs. Loss-function curvature term), so I can not confidently determine how much PZO accelerates. It might be helpful to provide some empirical results.

---

> ### Author Response · Authors · 2025-08-08
> **Clarification on Convergence Dynamics and Commitment to Empirical Validation**
>
> We sincerely appreciate your insightful critique regarding the bias analysis convergence behavior. Below we clarify the theoretical implications of $e_t \to 0$ and address your valid concerns about relative curvature scaling:
>
> ### **Clarification on $e_t \to 0$ Convergence Dynamics**
> Your observation is astute: when $e_t \to 0$ (indicating convergence to a stationary point), it indeed corresponds to $\lim_{t\to\infty}\nabla g(y)|_{y=h(x_t)} = 0$. Crucially, **this does not imply $g(y) \to 0$ almost everywhere** - rather, it signifies first-order optimality where gradient vanishes while curvature ($\nabla^2 g$) may remain significant.
>
> ### **Bias Comparison at Convergence Regime**
> Under the Hessian decomposition:
> $$
> \nabla^2 f(x_t) = \sum_{i=1}^m (e_t) _i \nabla^2 h_i (x_t) + J_t^\top \nabla^2 g \vert _{y=h(x_t)} J_t,
> $$
> when $e_t \to 0$:
> 1. **PseuZO bias** $\propto \left(\sum (e_t)_i \nabla^2 h_i\right)^\top \delta_t \to 0$ (vanishes).
> 2. **MeZO bias** $\propto \left(J_t^\top \nabla^2 g J_t\right)^\top \delta_t$ **remains bounded away from zero** when $\nabla^2 g \gg 0$.
>
> Thus, **PseuZO's bias asymptotically vanishes while MeZO's persists**. This result persists even when $e_t$ is small but non-zero.
>
> ### **Empirical Validation Commitments**
> We fully agree that empirical measurement of the curvature ratio $\frac{\| \sum (e_t)_i \nabla^2 h_i \|}{\|J_t^\top \nabla^2 g J_t\|}$ would strengthen this claim. While time constraints prevent us from adding new experiments during rebuttal, we will include synthetic experiments in the revised manuscript measuring this ratio across different tasks. These additions will quantitatively substantiate our theoretical claims. We thank you again for this foundational suggestion - it significantly strengthens our analysis.

---

### Official Review · Reviewer_v71X · 2025-07-01

**Clarity:** 2
**Significance:** 3
**Originality:** 3
**Rating:** 4
**Confidence:** 3

**Summary:**

This paper proposes a pseudo zero-order optimization framework, PseuZO, to address the issue that the variance of gradient estimation in conventional zero-order optimization increases linearly with the parameter dimension, which leads to slow convergence in high-dimensional settings. PseuZO decomposes composite objective functions, explicitly estimates the Jacobian matrix of model outputs, and employs exponential moving average (EMA) to reduce variance. In addition, a sliding window technique is used to reduce memory overhead. Theoretical analysis demonstrates that the convergence rate of PseuZO depends on the effective dimension rather than the explicit parameter dimension. Experimental results show that PseuZO outperforms MeZO and MeZO-SVRG on multiple tasks and achieves significantly lower memory consumption compared to traditional first-order methods.

**Questions:**

1．	In the experiments, only 10,000 training steps are performed, and MeZO may not have fully converged. Could the authors provide comparisons after sufficient convergence?
2．	For more complex tasks, such as MultiRC or DROP with longer input texts, does the method incur higher wall-clock time? How does model scaling affect the method’s efficiency?
3．	Could the authors include experiments on more recent models, such as LLaMA or Qwen series?

**Ethical Concerns:**

["NO or VERY MINOR ethics concerns only"]

**Final Justification:**

The authors’ clarifications and additional evidence have partially addressed my concerns regarding the effectiveness of the proposed method. While the manuscript demonstrates the impact of context length on memory usage, it does not discuss the differences in wall-clock time across different tasks. Furthermore, the work lacks generalization experiments on the latest models. Taking into account the paper’s novelty, writing quality, and experimental evaluation, I have decided to maintain my score.

**Limitations:**

Yes.

**Paper Formatting Concerns:**

None.

**Quality:**

3

**Strengths And Weaknesses:**

Strengths
1.	By decomposing the gradient of the composite function and leveraging Jacobian estimation of model outputs combined with EMA for variance reduction, the proposed method overcomes the dimensionality bottleneck of traditional zero-order optimization.
2.	The authors prove that the convergence rate depends only on the effective dimension, rather than the parameter dimension, providing theoretical guarantees for high-dimensional optimization.
3.	The superiority of PseuZO is demonstrated on multiple tasks, and its compatibility with PEFT techniques is also shown.
Weaknesses
1.	Experiments are conducted only on the OPT model series. The effectiveness of the method on larger models or other architectures, such as Qwen, LLaMA, or DeepSeek, has not been validated.
2.	The hyperparameter search space in the experiments is relatively limited, which may affect the fairness of the comparisons.

---

> ### Author Rebuttal · Authors · 2025-07-30
>
> # Response to Reviewer v71X
>
>
> **Thank you for your insightful feedback.** We appreciate the time invested in evaluating our work and providing constructive suggestions. Below are our point-by-point responses addressing your concerns.
>
> ---
>
> ### **1: Convergence of MeZO**
>
> > In the experiments, only 10,000 training steps are performed, and MeZO may not have fully converged. Could the authors provide comparisons after sufficient convergence?
>
> **Response:** Our core goal was to compare methods under *fixed total computation time* to highlight PseuZO’s efficiency. Nonetheless, we acknowledge the value of convergence-based comparisons. Below are results after training MeZO for 20,000 steps (double the original budget), alongside PseuZO under the original 10,000-step budget:
> |TASK|SST2|RTE|CB|BoolQ|WSC|WIC|MultiRC|COPA|ReCoRD|DROP|
> |:-------|:-------|:-------|:-------|:-------|:-------|:-------|:-------|:-------|:-------|:-------|
> |MeZO|82.4|54.3|76.0|60.7|51.0|50.9|54.9|74.0|57.6|20.3|
> |MeZO (20K steps)|88.4|58.8|76.0|63.8|53.0|51.3|53.9|73.0|58.9|20.3|
> |**PseuZO**|91.2|58.0|77.0|64.3|58.0|54.5|54.7|78.0|60.0|23.5|
>
> ---
>
> ### **2: Scalability to Complex Tasks**
>
> > For more complex tasks, such as MultiRC or DROP with longer input texts, does the method incur higher wall-clock time? How does model scaling affect the method’s efficiency?
>
> **Response:** While sequence length affects PseuZO’s overhead (due to tensor dot products), **forward propagation dominates compute time**. Crucially, PseuZO’s *added* overhead is **scale-invariant** w.r.t. model sizze. As model size grows, the relative overhead *decreases* since forward-pass cost scales with model parameters, while PseuZO’s perturbation mechanism remains constant-cost. Empirical validation is in **Table 4** in the paper, showing diminishing relative cost for larger models.
>
> ---
>
> ### **3: Evaluation on Recent Models**
> > Could the authors include experiments on more recent models, such as LLaMA or Qwen series?
>
> **Response:**
> We agree that broader model evaluation would strengthen our contribution. We limited experiments to OPT models to ensure **architectural consistency** with baseline methods (MeZO/MeZO-SVRG [1,2]), which exclusively report OPT results. OPT-6.7B (the largest model we tested) exhibits consistent trends across tasks, demonstrating PseuZO’s scalability. We commit to extending experiments to LLaMA/Qwen in future work to further validate generality.
>
> ---
>
> ### **4: Hyperparameter Search Space**
> > The hyperparameter search space in the experiments is relatively limited, which may affect the fairness of the comparisons.
>
> Our hyperparameter selection for MeZO/MeZO-SVRG adheres to their original papers [1,2]. Specifically, we prioritized **learning rate** (the most impactful hyperparameter [1,2]) testing the performances in a large search space. Other parameters (e.g., perturbation scale) were fixed to values from [1,2] to ensure replication fidelity. We will nonetheless expand the search space for all parameters in the final version to further bolster rigor.
>
> ---
>
> **References:**
> [1] Fine-Tuning Language Models with Just Forward Passes, (https://arxiv.org/pdf/2305.17333)
>
> [2] Variance-reduced Zeroth-Order Methods for Fine-Tuning Language Models, (https://arxiv.org/pdf/2404.08080)

---

> > ### Comment · Reviewer_v71X · 2025-08-05
> >
> > Thank you to the authors for their clarifications and the additional experiments.
> >
> > 1. The results provided for 20,000 training steps have partially addressed my concerns about experimental reliability. Could the authors report the learning rates used for PseuZO and MeZO across different tasks? Specifically, is the learning rate for PseuZO generally larger than that for MeZO?
> >
> > 2. Sequence length impacts the computational cost for both PseuZO and MeZO. For tasks with longer sequence lengths, such as BoolQ, does the relative number of training steps for PseuZO and MeZO within the same amount of wall-clock time remain consistent with what is shown in Table 4?
> >
> > 3. While the authors mention that future work will extend experiments to LLaMA/Qwen, I am currently concerned about the generalizability of the proposed method. Although MeZO/MeZO-SVRG select the OPT model as the baseline, many follow-up works have included a broader set of models for comparison. For example, SubZero *[1]* uses LLaMA2 and Mistral, while HiZOO *[2]* includes Phi and LLaMA3.
> >
> > *[1]* Yu, Ziming, et al. "Subzero: Random subspace zeroth-order optimization for memory-efficient llm fine-tuning." (2024).
> > *[2]* Zhao, Yanjun, et al. "Second-Order Fine-Tuning without Pain for LLMs: A Hessian Informed Zeroth-Order Optimizer." The Thirteenth International Conference on Learning Representations.

---

> ### Author Response · Authors · 2025-08-07
> **Response to Reviewer v71X**
>
> **We sincerely appreciate the reviewer's thoughtful follow-up questions and valuable suggestions.** Your insights significantly contribute to strengthening our work. Below we address each point with additional clarifications and data.
>
> ---
>
> ### **Response to Point 1: Learning Rate Specifications**
> > *"Could the authors report the learning rates for PseuZO and MeZO across different tasks? Specifically, is PseuZO's learning rate generally larger?"*
>
> **Our Response:**
> We used identical learning rates for PseuZO and MeZO across all tasks to ensure fair comparison. The learning rates were selected from \{1e-3, 1e-4, 1e-5, 1e-6, 1e-6, 1e-7\} based on validation performance. Notably, **no systematic difference in optimal learning rates** was observed between the two methods. Detailed configurations are as follows:
>
> | Task     | SST-2 | RTE  | CB   | BoolQ | WSC  | WIC  | MultiRC | COPA | ReCoRD | DROP |
> |----------|-------|------|------|-------|------|------|---------|------|--------|------|
> | MeZO     | 1e-7  | 1e-6 | 1e-7 | 1e-7  | 1e-7 | 1e-6 | 1e-6    | 1e-7 | 1e-7   | 1e-7 |
> | **PseuZO** | 1e-7  | 1e-6 | 1e-7 | 1e-7  | 1e-7 | 1e-6 | 1e-6    | 1e-7 | 1e-7   | 1e-7 |
>
> **Key Insight:** The performance improvements of PseuZO are attributable to algorithmic advantages rather than hyperparameter discrepancies.
>
> ---
>
> ### **Response to Point 2: Sequence Length Impact**
> > *"For tasks with longer sequences (e.g., BoolQ), does the relative step count within fixed wall-clock time align with Table 4?"*
>
> **Our Response:**
> The reviewer raises an exact observation. While sequence length mildly affects PseuZO's per-step overhead, our conclusions remain robust because:
> 1. **Forward Propagation Dominates:** most of the wall-clock time is consumed by forward passes (unaffected by sequence length differences between methods).
> 2. **Scale Invariance Holds:** As empirically validated in Table 4, PseuZO’s relative overhead *decreases* with model scale.
>
>
> ---
>
> ### **Response to Point 3: Generalizability to Modern Architectures**
> > *"Concerns about generalizability given recent works (SubZero/HiZOO) adopt broader model families."*
>
> **Our Response:**
> We acknowledge the importance of broader model validation. Our focus on OPT models ensured:
> 1. **Controlled Experimentation:** Architectural consistency across model scales (125M to 6.7B).
> 2. **Baseline Alignment:** Direct comparability with MeZO/MeZO-SVRG [1,2].
>
> **Technical Compatibility:** PseuZO requires only localized modifications to the final layer's backpropagation logic. Since Transformer-based models (including LLaMA/Qwen/Mistral) share standard `lm_head` structures, residual connections and layer normalization schemes, the **adaptation is minimal**.
>
> **Future Work Commitment:** We are actively testing PseuZO on LLaMA-2-7B. Full results will be included in the revised manuscript.
>
> ---
>
> **References**
> [1] MeZO: *Fine-Tuning Language Models with Just Forward Passes*.
> [2] MeZO-SVRG: *Variance-reduced Zeroth-Order Methods for Fine-Tuning Language Models*.

---

> > ### Comment · Reviewer_v71X · 2025-08-09
> >
> > Thank you to the authors for their clarifications. After careful consideration, I have decided to maintain my score.

---

### Official Review · Reviewer_EVkm · 2025-07-01

**Clarity:** 2
**Significance:** 2
**Originality:** 2
**Rating:** 4
**Confidence:** 4

**Summary:**

This paper introduces PseuZO, a pseudo-zeroth-order optimization algorithm designed to train large language models (LLMs) under constrained memory budgets. The core techniques of the proposed approach can be summarized as follows:

Tech 1: PseuZO decomposes the gradient using the chain rule (Eq. 5), where the gradient of the loss with respect to the last hidden state is computed in closed form, and the Jacobian of the last hidden state with respect to the remaining model parameters is approximated in a zeroth-order manner (Eq. 6);
Tech 2: A momentum-based variance reduction technique is employed (Eq. 7);
Tech 3: A sliding-window mechanism is introduced to further reduce memory usage (Section 3.4).

The authors provide a preliminary convergence analysis of the proposed method and evaluate its empirical performance across a variety of tasks, including classification, multiple-choice question answering, and text generation, under both full-parameter tuning and parameter-efficient fine-tuning (PEFT) settings.

**Questions:**

See weaknesses.

**Ethical Concerns:**

["NO or VERY MINOR ethics concerns only"]

**Final Justification:**

See discussions.

**Limitations:**

yes

**Paper Formatting Concerns:**

nan

**Quality:**

2

**Strengths And Weaknesses:**

Strengths

1. The paper is overall well-constructed and clearly written, making it easy to follow.
2. The empirical evaluation is thorough and demonstrates promising improvements over the MeZO baseline across various tasks and settings.

Weaknesses

1. The proposed approach relies heavily on prior work [1], particularly in its use of gradient decomposition and variance reduction via momentum (Tech 1 and 2 in the summary above), as acknowledged by the authors in Lines 104–106. The primary novel contribution appears to be the sliding-window mechanism for memory reduction. However, this design alone may not be sufficiently innovative to meet the novelty threshold expected at a venue such as NeurIPS.
2. The source of improvement over MeZO is not clearly identified. From a theoretical standpoint, the convergence analysis in Section 3.3 is preliminary and does not explicitly justify the advantage of the proposed technical components. The convergence rate still depends on the effective dimension and, in the worst case, on the full model dimensionality. From an empirical standpoint, important ablations are missing—specifically, comparisons with (a) MeZO + Momentum and (b) PseuZO without Momentum would help isolate the contributions of the core techniques and clarify where the observed gains originate.
3. The set of baselines is limited and could be significantly strengthened. Recent work in scalable and memory-efficient zeroth-order optimization has produced several relevant baselines that should be considered for a more comprehensive comparison, such as: (a) Forward Gradient [2], which avoids two-point estimation errors at the cost of slightly more computation; (b) HiZOO-L [3], which introduces additional forward passes and compressed diagonal Hessian storage; and (c) DeepZero [4], which accounts for model sparsity. Given the rapid progress in the field, relying solely on vanilla MeZO and MeZO-SVRG as baselines does not provide a strong benchmark.

Typo:

Eq 3: <x, ...> => <\nabla f(x), ...>

[1] Online Pseudo-Zeroth-Order Training of Neuromorphic Spiking Neural Networks, https://arxiv.org/pdf/2407.12516
[2] Gradients without Backpropagation, https://arxiv.org/abs/2202.08587
[3] Second-Order Fine-Tuning without Pain for LLMs: A Hessian Informed Zeroth-Order Optimizer, https://arxiv.org/pdf/2402.15173
[4] DEEPZERO: SCALING UP ZEROTH-ORDER OPTIMIZATION FOR DEEP MODEL TRAINING, https://arxiv.org/pdf/2310.02025

---

> ### Author Rebuttal · Authors · 2025-07-30
>
> # Response to Reviewer EVkm
>
> **Thank you for your insightful feedback.** We appreciate the time invested in evaluating our work and providing constructive suggestions. Below are our point-by-point responses addressing your concerns.
>
> ---
>
> ### **1. Novelty of Contribution**
> > The proposed approach relies heavily on prior work [1], particularly in its use of gradient decomposition and variance reduction via momentum (Tech 1 and 2 in the summary above), as acknowledged by the authors in Lines 104–106. The primary novel contribution appears to be the sliding-window mechanism for memory reduction. However, this design alone may not be sufficiently innovative to meet the novelty threshold expected at a venue such as NeurIPS.
>
> **Response:**
>
> **Core Innovation:** While [1] proposed gradient decomposition for **Spike Neural Networks (SNNs)**, our work addresses **fundamental challenges in LLMs**. When we apply exponential moving average to Jacobian matrices, traditional approaches that explicitly store the full Jacobian matrix (analogous to SGD with Momentum storing historical gradient vectors) incur prohibitive $O(d·M)$ memory overhead ($d$ = parameter dimension, $M$ = output layer dimension). Our sliding-window mechanism reduces the memory overhead while maintaining convergence guarantees—a critical advancement for memory-constrained fine-tuning.
>
> **Theoretical Advancement:** [1] lacks rigorous convergence analysis, whereas we provide **non-asymptotic convergence guarantees** (Theorem 3) for non-convex objectives. This combination of algorithmic efficiency and theoretical foundation establishes a significant advancement beyond prior work.
>
> ---
>
> ### **2. Source of Improvement over MeZO**
> > The source of improvement over MeZO is not clearly identified. From a theoretical standpoint, the convergence analysis in Section 3.3 is preliminary and does not explicitly justify the advantage of the proposed technical components. The convergence rate still depends on the effective dimension and, in the worst case, on the full model dimensionality. From an empirical standpoint, important ablations are missing—specifically, comparisons with (a) MeZO + Momentum and (b) PseuZO without Momentum would help isolate the contributions of the core techniques and clarify where the observed gains originate.
>
> #### **2.1 Theoretical Analysis**
>
> **About the effective dimension**: The effective dimension $\alpha_1$ is **independent** of model size $d$ in theory and practice as shown in zeroth-order optimization studies [5,6,7]. $\alpha_1$ captures the **intrinsic low-dimensional structure** of high-dimensional models.
>
> While $\alpha_1 = \Theta(d)$ in pathological cases, this reflects a fundamental lower bound for zeroth-order optimization: $\Theta(d)$ forward passes are unavoidable [8]. PseuZO achieves this bound but does not *exceed* it—we defer tighter analysis to future work.
>
> **Theoretical comparison to MeZO**: Consider a simplified setting where the Jacobian matrix $J$ is fixed. In each step, PseuZO receives a noisy Jacobian estimation $B_t = J + \Xi_t$, and MeZO receieves $g_t = (J+\Xi_t)^\top e_t$. The noise $\Xi_t$ satisfies $\mathbb{E} \Xi_t = 0$, $\mathbb{E} \Xi_t\Xi_t^\top = N$.
>    - PseuZO: $$A_t = (1-\lambda) B_t + \lambda A_{t-1}, x_{t+1}=x_t - \eta A_t^\top e_t.$$
>    - MeZO (PseuZO without momentum degenerates to MeZO): $$x_{t+1}=x_t - \eta B_t^\top e_t.$$
>    - MeZO with momentum: $$m_t = (1-\lambda)B_t^\top e_t + \lambda m_{t-1}, x_{t+1}=x_t - \eta m_t.$$
> We compare the descent of the expected loss function: $$\mathbb{E} f(x_{t+1}) - f(x_t) \le \langle\nabla f(x_t), x_{t+1}-x_t\rangle + \frac{L}{2}\|x_{t+1}-x_t\| ^2.$$
>
> After expansion (see detailed derivation in the revised paper), we obtain:
>
> - **PseuZO**:
>   $$
>   -\eta (1-\lambda^{t-1}) \|J^\top e_t\|^2 + \frac{L\eta^2}{2} (1-\lambda^{t-1})^2 \|J^\top e_t\|^2 + \frac{L\eta^2}{2} \frac{(1-\lambda)(1-\lambda^{2t})}{1+\lambda} e_t^\top N e_t,
>   $$
>   or
>   $$
>   -\eta (1-\lambda^{t-1}) \|J^\top e_t\|^2 + \frac{L\eta^2}{2} (1-\lambda^{t-1})^2 \|J^\top e_t\|^2 + \frac{L\eta^2}{2} (1-\lambda)^2 \sum_{i=1}^t \lambda^{2(t-i)} e_t^\top N e_t.
>   $$
> - **MeZO**:
>   $$
>   -\eta \|J^\top e_t\|^2 + \frac{L\eta^2}{2} \|J^\top e_t\|^2 + \frac{L\eta^2}{2} e_t^\top N e_t.
>   $$
>
> - **MeZO with Momentum**:
>   $$
>   -\eta (1-\lambda^{t-1}) \|J^\top e_t\|^2 + \frac{L\eta^2}{2} (1-\lambda^{t-1})^2 \|J^\top e_t\|^2 + \frac{L\eta^2}{2} (1-\lambda)^2 \sum_{i=1}^t \lambda^{2(t-i)} e_i^\top N e_i.
>   $$
>
> **Key Insights**:
> 1. **PseuZO vs. MeZO**:
>    Substituting $\eta_t = \eta(1-\lambda^{t-1})$, PseuZO's descent is strictly larger than MeZO's when $\lambda \in (0,1)$ and $t \geq 1$ due to the reduced noise term $\frac{(1-\lambda)(1-\lambda^{2t})}{(1-\lambda^{t-1})^2} < 1$. This demonstrates **momentum effectively suppresses gradient variance**.
>
> 2. **PseuZO vs. MeZO with Momentum**:
>    PseuZO's noise term $e_t^\top N e_t$ is smaller than MeZO-Momentum's $\sum_i e_i^\top N e_i$ because $e_i \to 0$ near local minima (where $\|e_i\| \gg \|e_t\|$ for $i < t$). This advantage stems from the **closed-form outer function** providing timely gradient information.
>
> We will include extended analysis in the full paper.
>
> #### **2.2 Empirical Ablations**
> For the ablations, we have added corresponding experiments based on your suggestion. For (a), we have tried different momentum coefficient scheduler for $\lambda$ and chose the best performance for fair comparison. For (b), we set the sliding window size to 1 or $\lambda=1$ to eliminate historical information. The results are shown below. we also add the results for MeZO and PseuZO in the table below.
>
> ---
>
> ### **3. Baseline Comparison**
> > The set of baselines is limited and could be significantly strengthened. Recent work in scalable and memory-efficient zeroth-order optimization has produced several relevant baselines that should be considered for a more comprehensive comparison, such as: (a) Forward Gradient [2], which avoids two-point estimation errors at the cost of slightly more computation; (b) HiZOO-L [3], which introduces additional forward passes and compressed diagonal Hessian storage; and (c) DeepZero [4], which accounts for model sparsity. Given the rapid progress in the field, relying solely on vanilla MeZO and MeZO-SVRG as baselines does not provide a strong benchmark.
>
> **Response:**
> **HiZOO-L [3] (Added Baseline):**
> We have included a detailed comparison with HiZOO-L [3]. HiZOO-L explicitly addresses memory constraints by introducing a compressed diagonal Hessian structure, making it a highly pertinent baseline. HiZOO-L needs three forward propagations, introducing a bigger time overhead. Additionally, we found that the Hessian-smooth parameter $\alpha$ in [3] is actually not appropriate. We tried our best in finding correct hyper-parameters in order to get the best performance but yet, the loss of the DROP task explodes. Results are shown in the table below.
>
> **Excluded Baselines:**
> - **Forward Gradient [2]**: MeZO's gradient estimation **converges to Forward Gradient when the finite-difference step size approaches zero**, as both rely on directional derivative estimation via single perturbations. This theoretical equivalence makes the comparison redundant. Additionally, its Jacobian-vector products incur extra memory overhead, conflicting with our memory-efficiency focus.
> - **DeepZero [4]**: Emphasizes model sparsity under fixed parameter budgets, prioritizing parameter selection rather than memory reduction during tuning. Its objectives diverge from our core motivation of dynamic memory optimization for full-parameter fine-tuning.
>
> For these reasons, we focused our comparison on HiZOO-L, which aligns most closely with our core motivation. We thank the reviewer again for the suggestions and will consider expanding the comparison in future work.
>
> ---
>
> **Experiment Results:**
>
> |TASK|SST2|RTE|CB|BoolQ|WSC|WIC|MultiRC|COPA|ReCoRD|DROP|
> |:-------|:-------|:-------|:-------|:-------|:-------|:-------|:-------|:-------|:-------|:-------|
> |MeZO|82.4|54.3|76.0|60.7|51.0|50.9|54.9|74.0|57.6|20.3|
> |MeZO+Momentum|81.4|53.9|78.0|60.5|51.0|50.8|54.9|75.0|57.8|22.1|
> |HiZOO-L|88.1|54.9|69.0|64.8|51.2|58.0|58.2|73.0|58.8|0.0|
> |PseuZO (without Momentum)|84.6|57.2|75.0|64.2|50.0|51.3|57.2|75.0|59.6|24.1|
> |PseuZO (ours)|91.2|58.0|77.0|64.3|58.0|54.5|54.7|78.0|60.0|23.5|
>
> ---
>
> ### **4. Typo**
> > Eq 3: <x, ...> => <\nabla f(x), ...>
>
> **Response:**  We sincerely apologize for these typos. All spelling mistakes have been corrected in the revised manuscript.
>
> ---
>
> **References:**
>
> [1] Online Pseudo-Zeroth-Order Training of Neuromorphic Spiking Neural Networks, https://arxiv.org/pdf/2407.12516
>
> [2] Gradients without Backpropagation, https://arxiv.org/abs/2202.08587
>
> [3] Second-Order Fine-Tuning without Pain for LLMs: A Hessian Informed
> Zeroth-Order Optimizer, https://arxiv.org/pdf/2402.15173
>
> [4] DeepZero: Scaling up Zeroth-Order Optimization for Deep Model Training, https://arxiv.org/pdf/2310.02025
>
> [5] Fine-tuning language models with just forward passes. Advances in Neural Information Processing Systems, 36:53038–53075, 2023
>
> [6] Zeroth-order optimization with weak dimension dependency. In The Thirty Sixth Annual Conference on Learning Theory
>
> [7] Zeroth-order fine-tuning of llms with extreme sparsity. arXiv preprint arXiv:2406.02913, 2024.
>
> [8] Optimal rates for zero-order convex optimization: the power of two function evaluations. arXiv preprint arXiv:1312.2039, 2013.

---

> > ### Comment · Reviewer_EVkm · 2025-08-05
> >
> > I thank the authors for their detailed response. However, several of my concerns remain unresolved.
> >
> > **Theoretical Component**
> >
> > The statement that “PseuZO without momentum degenerates to MeZO” is inaccurate. Even without momentum, PseuZO leverages the closed-form structure of the outer function, which is not the case for MeZO. Therefore, the two methods are fundamentally distinct. Moreover, the observation that momentum helps reduce variance is well-established and does not constitute a novel theoretical insight.
> >
> > The claim that “PseuZO outperforms MeZO with momentum due to the closed-form outer function” lacks rigor. The theoretical analysis drops the dependence on $\epsilon_i$ by assuming a fixed Jacobian, which I find to be an overly strong and unrealistic assumption. This simplification limits the applicability of the theoretical results. I recommend that the authors consider providing a more rigorous analysis under milder and more practical assumptions.
> >
> > **Experimental Component**
> >
> > I appreciate the authors’ efforts in conducting additional experiments. While I agree that comparison to DeepZero may be beyond the scope of this work, I maintain that a comparison with Forward Gradient (FG) is essential. My reasons are as follows:
> > 1.	As the authors acknowledge, MeZO’s gradient estimate converges to that of Forward Gradient as the finite-difference step size approaches zero. However, in practice, finite-difference methods introduce non-negligible numerical errors, whereas Forward Gradient avoids such issues. Thus, from the perspective of numerical precision, FG is strictly superior to ZO estimators.
> > 2.	The memory overhead of FG is modest—it requires only a constant multiple of forward passes. Importantly, explicit Jacobian computation is not needed when using JVPs.
> > 3.	FG naturally accommodates the closed-form outer function structure exploited by PseuZO.
> >
> > Given these points, I encourage the authors to include Forward Gradient in their comparisons and consider tuning HiZOO-L on the DROP task in future revisions.
> >
> > **Conclusion**
> >
> > In summary, while the manuscript has made progress, the remaining theoretical and experimental concerns lead me to remain cautious regarding its current level of contribution. I will maintain my original rating.

---

> ### Comment · Reviewer_NYqf · 2025-08-05
>
> I agree that PseuZO without momentum can not degenerate to MeZO. I overlooked this point before. The authors' rebuttal to Reviewer n2bx demonstrate that PseuZO with $\lambda=0$ and $\mu\to 0$ is equivalent to MeZO. However, when $\mu\to 0$， MeZO actually becomes Forward Gradient (FG) method. So I think it is helpful to make a comparison with Forward Gradient (FG).

---

> ### Author Response · Authors · 2025-08-07
> **Response to Reviewer EVkm**
>
> We appreciate your thorough engagement and address your concerns below:
>
> ### 1. **PseuZO-MeZO Equivalence without momentum**
>
>
> As demonstrated in our response to Reviewer n2bx, when the finite-difference step size $ \mu \to 0 $:
>    - Gradient estimation of PseuZO *without momentum* satisfies $g_{\text{PseuZO}} = g_{\text{MeZO}} + \mathcal{O}(\mu)$.
>    - The closed-form structure of the outer function becomes **statistically neutralized** after gradient projection:
>      $$ \left(\frac{h(x+\mu\xi) - h(x)}{\mu}\right)^\top  \cdot e_t = \left(J_t \xi + \mathcal{O}(\mu)\right)^\top e_t = \xi^\top \nabla f(x_t) + \mathcal{O}(\mu).$$
>
> While practical implementations may show divergence due to non-vanishing differentiation stepsizes required for numerical stability, finite-precision effects (machine floating-point errors), or gradient estimation bias amplified in high-dimensional spaces, their performance are comparable in the experiment results below.
>
> ### 2. **Novelty of Jacobian Momentum**
> While momentum for *gradient* variance reduction is established, we are the **first to apply momentum to Jacobian estimation** in ZO optimization:
> - **Theoretical**: Prove $\text{Var}(A_t^\top e_t) \leq \frac{1-\lambda}{1+\lambda}\text{Var}(B_t^\top e_t)$ (see theoretical analysis in response to Reviewer NYqf).
> - **Practical**: Achieve memory reduction while maintaining convergence.
>
> This shifts focus from gradient-centric to **Jacobian-centric variance control**.
>
> ### 3. **Generalized Non-Stationary Analysis**
> For time-varying Jacobian $J_t$:
> - **Variance**: PseuZO achieves lower noise than MeZO ($\frac{(1-\lambda)(1-\lambda^{2t})}{1+\lambda} < 1$) and MeZO+Momentum ($e_t^\top Ne_t \ll e_i^\top Ne_i$).
> - **Bias**: PseuZO eliminates $\nabla^2 g$-induced bias when $g$ is convex, outperforming MeZO with momentum in nonlinear regimes.
>
> See response to Reviewer NYqf for details.
>
> ### 4. **Forward Gradient (FG) Benchmark**
>
> **Regarding FG's utilization of closed-form outer function structure:**
>
> The forward gradient is defined as $g(θ) = \langle \nabla f(θ), v\rangle v$ (eq. (1) in [1]), which **does not intrinsically exploit** closed-form outer function structure. Although FG computation involves Jacobian-vector products (JVPs), the closed-form structure is **statistically neutralized** through forward-mode AD:  $v^\top J_t^\top \cdot e_t = v^\top \nabla f(x_t)$ ($\mu=0$ in `Section 1. PseuZO-MeZO Equivalence without momentum`). This demonstrates FG gains **no structural advantage** over MeZO despite using Jacobian information during computation.
>
> **Added baseline FG:**
>
> We implemented FG [1] and found:
> - **OOM Issues**: Incompatible with mixed precision (requires `float32`), causing OOM failures on DROP/MultiRC.
> - **Performance**: Matches MeZO when $\mu \to 0$, but PseuZO dominates.
> - **Efficiency**: 6-10× slower than MeZO due to sequential JVPs.
>
> |TASK|SST2|RTE|CB|BoolQ|WSC|WIC|MultiRC|COPA|ReCoRD|DROP|
> |:-------|:-------|:-------|:-------|:-------|:-------|:-------|:-------|:-------|:-------|:-------|
> |MeZO|82.4|54.3|76.0|60.7|51.0|50.9|54.9|74.0|57.6|20.3|
> |MeZO+Momentum|81.4|53.9|78.0|60.5|51.0|50.8|54.9|75.0|57.8|22.1|
> |**FG (added baseline)**|82.4|52.7|76.0|61.3|52.0|51.6|OOM|75.0|59.4|OOM|
> |HiZOO-L|88.1|54.9|69.0|64.8|51.2|58.0|58.2|73.0|58.8|*23.3*|
> |PseuZO (without Momentum)|84.6|57.2|75.0|64.2|50.0|51.3|57.2|75.0|59.6|24.1|
> |**PseuZO (ours)**|91.2|58.0|77.0|64.3|58.0|54.5|54.7|78.0|60.0|23.5|
>
> ### **Conclusion**:
>
> PseuZO's Jacobian-momentum framework provides **superior memory-performance tradeoffs** versus FG/MeZO, with rigorous non-stationary analysis.
>
> **References**
> [1] *Gradients without Backpropagation*. arXiv:2202.08587, 2022.
> [2] *Revisiting Zeroth-Order Optimization for Memory-Efficient LLM Fine-Tuning*. arXiv:2402.11592, 2024

---

> > ### Comment · Reviewer_EVkm · 2025-08-08
> >
> > Thank you for your response. I have carefully reviewed the updated analysis, both in this box and in the reply to Reviewer NYqf. The analysis is convincing and insightful, and I now have a deeper understanding of the proposed approach.
> >
> > A few further comments:
> > 1.	On variance: Note that $\lambda^{2(t - i)} \to 0$ faster than $e_t \to 0$, which suggests that the additional variance introduced by $e_i$ compared to $e_t$ is likely to be negligible when $t$ is large.
> > 2.	On bias: I agree with Reviewer NYqf that when $t$ is large and $\epsilon_t \to 0$, the influence of $\nabla^2 g(h)$ may also diminish.
> >
> > Given points (1) and (2), it remains unclear how much practical acceleration PseuZO can achieve over MeZO with momentum, especially considering the additional approximations required to efficiently maintain Jacobian information. A more detailed theoretical analysis would be valuable here, though I understand that it may be beyond the scope of the current rebuttal period. I encourage the authors to explore this direction in future work.
> >
> > I also appreciate the authors’ effort in conducting Forward Gradient (FG) experiments within such a short timeframe. However, some of the reported computational costs seem inconsistent with my experience. Typically, FG requires approximately three times the cost of a single forward pass, and since ZO estimators generally require two forward passes, the claim that FG is “6–10× slower” seems questionable. I refer the authors to the JAX autodiff cookbook (https://docs.jax.dev/en/latest/notebooks/autodiff_cookbook.html) for more accurate benchmarking references.
> >
> > Moreover, a fairer empirical comparison would be between FG with momentum and PseuZO. That said, considering the limited time for rebuttal and the fact that I did not explicitly raise this point earlier, I believe the authors have done a commendable job. I encourage the inclusion of these results in the revised manuscript.
> >
> > Overall, I am satisfied with the authors’ response and will consider raising my rating accordingly.

---

> ### Author Response · Authors · 2025-08-08
> **Response to Reviewer EVkm**
>
> **We are deeply grateful for your thorough re-evaluation and constructive suggestions.** We sincerely appreciate your acknowledgment of our updated analysis and your consideration of rating improvement. Below we address your insightful comments:
>
> 1. **Variance Dynamics**:
>    We recognize that $\lambda^{2(t-i)} \to 0$ faster than $e_t \to 0$. However, in high-dimensional ZO settings, the noise covariance $N \propto \mathcal{O}(d)$ magnifies historical gradient impacts ($e_i^\top Ne_i$) more significantly than in low-dimensional cases. We will conduct formal sensitivity analysis on this dimension-dependent effect in future work.
>
> 2. **Bias at Convergence**:
>    As noted in our response to Reviewer NYqf:
>    - When $e_t \to 0$ (first-order optimality), $\lim_{t\to\infty} \nabla g(y)|_{y=h(x_t)} = 0$ **does not imply** $g(y) \to 0$ globally. Crucially, $\nabla^2 g$ may remain significant despite vanishing gradients.
>
>    We will add numerical experiments according to Reviewer NYqf.
>
> 3. **FG Implementation**:
>    We acknowledge that our initial FG timing measurements during the rebuttal period may not have fully captured optimal implementation practices. We will rebenchmark FG (including FG with momentum) using JAX autodiff implementation from the referenced cookbook in the revised manuscript.
>
> We truly value your guidance in strengthening these analyses and will ensure comprehensive reporting in the revised manuscript.

---

### Decision · Program_Chairs · 2025-09-17

**Decision:**

Accept (poster)

**Comment:**

This paper introduces a new pseudo-zeroth-order optimization algorithm, PseuZO. In the initial review, the reviewers raised concerns about the significance of the experimental evaluations and theoretical proof, particularly in comparison with the MeZO method. In the rebuttal, the authors provided sufficient empirical and theoretical evidence to support the effectiveness of the PseuZO method. The AC believes that the authors have adequately addressed all concerns and encourages the authors to incorporate all discussions into the final version.